# Deep kernel processes

## Abstract

We define deep kernel processes in which positive definite Gram matrices are progressively transformed by nonlinear kernel functions and by sampling from (inverse) Wishart distributions. Remarkably, we find that deep Gaussian processes (DGPs), Bayesian neural networks (BNNs), infinite BNNs, and infinite BNNs with bottlenecks can all be written as deep kernel processes. For DGPs the equivalence arises because the Gram matrix formed by the inner product of features is Wishart distributed, and as we show, standard isotropic kernels can be written entirely in terms of this Gram matrix — we do not need knowledge of the underlying features. We define a tractable deep kernel process, the deep inverse Wishart process, and give a doubly-stochastic inducing-point variational inference scheme that operates on the Gram matrices, not on the features, as in DGPs. We show that the deep inverse Wishart process gives superior performance to DGPs and infinite BNNs on standard fully-connected baselines.

## 1 Introduction

The deep learning revolution has shown us that effective performance on difficult tasks such as image classification (Krizhevsky et al., 2012) requires deep models with flexible lower-layers that learn task-dependent representations. Here, we consider whether these insights from the neural network literature can be applied to purely kernel-based methods. (Note that we do not consider deep Gaussian processes or DGPs to be "fully kernel-based" as they use a feature-based representation in intermediate layers).

Importantly, deep kernel methods (e.g. Cho & Saul, 2009) already exist. In these methods, which are closely related to infinite Bayesian neural networks (Lee et al., 2017; Matthews et al., 2018; Garriga-Alonso et al., 2018; Novak et al., 2018), we take an initial kernel (usually the dot product of the input features) and perform a series of deterministic, parameter-free transformations to obtain an output kernel that we use in e.g. a support vector machine or Gaussian process. However, the deterministic, parameter-free nature of the transformation from input to output kernel means that they lack the capability to learn a top-layer representation, which is believed to be crucial for the effectiveness of deep methods (Aitchison, 2019).

To obtain the flexibility necessary to learn a task-dependent representation, we propose deep kernel processes (DKPs), which combine nonlinear transformations of the kernel, as in Cho & Saul (2009) with a flexible learned representation by exploiting a Wishart or inverse Wishart process (Dawid, 1981; Shah et al., 2014). We find that models ranging from DGPs (Damianou & Lawrence, 2013; Salimbeni & Deisenroth, 2017) to Bayesian neural networks (BNNs; Blundell et al., 2015, App. C.1), infinite BNNs (App. C.2) and infinite BNNs with bottlenecks (App. C.3) can be written as DKPs (i.e. only with kernel/Gram matrices, without needing features or weights). Practically, we find that the deep inverse Wishart process (DIWP), admits convenient forms for variational approximate posteriors, and we give a novel scheme for doubly-stochastic variational inference (DSVI) with inducing points purely in the kernel domain (as opposed to Salimbeni & Deisenroth, 2017, who described DSVI for standard feature-based DGPs), and demonstrate improved performance with carefully matched models on fully-connected benchmark datasets.

## 2 Background

We briefly revise Wishart and inverse Wishart distributions. The Wishart distribution is a generalization of the gamma distribution that is defined over positive semidefinite matrices. Suppose that

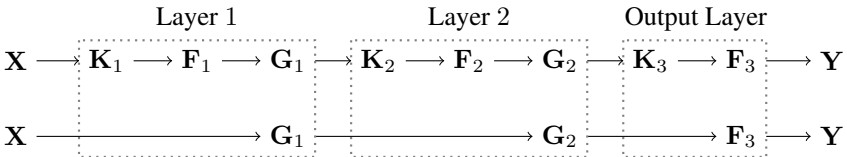

Figure 1: Generative models for two layer ($L = 2$) deep GPs. (**Top**) Generative model for a deep GP, with a kernel that depends on the Gram matrix, and with Gaussian-distributed features. (**Bottom**) Integrating out the features, the Gram matrices become Wishart distributed.

we have a collection of $P$-dimensional random variables $\mathbf{x}_i$ with $i \in \{1, \ldots, N\}$ such that

$$\mathbf{x}_i \overset{\text{iid}}{\sim} \mathcal{N}\left(\mathbf{0}, \mathbf{V}\right), \qquad \text{then,} \qquad \sum_{i=1}^{N}\mathbf{x}_i\mathbf{x}_i^T = \mathbf{S} \sim \mathcal{W}\left(\mathbf{V}, N\right) \qquad (1)$$

has Wishart distribution with scale matrix $\mathbf{V}$ and $N$ degrees of freedom. When $N > P - 1$, the density is,

$$\mathcal{W}\left(\mathbf{S}; \mathbf{V}, N\right) = \frac{1}{2^{NP}|\mathbf{V}|\Gamma_P\left(\frac{N}{2}\right)}|\mathbf{S}|^{(N-P-1)/2}\exp\left(-\tfrac{1}{2}\operatorname{Tr}\left(\mathbf{V}^{-1}\mathbf{S}\right)\right), \qquad (2)$$

where $\Gamma_P$ is the multivariate gamma function. Further, the inverse, $\mathbf{S}^{-1}$ has inverse Wishart distribution, $\mathcal{W}^{-1}\left(\mathbf{V}^{-1}, N\right)$. The inverse Wishart is defined only for $N > P - 1$ and also has closed-form density. Finally, we note that the Wishart distribution has mean $N\mathbf{V}$ while the inverse Wishart has mean $\mathbf{V}^{-1}/(N - P - 1)$ (for $N > P + 1$).

## 3 DEEP KERNEL PROCESSES

We define a kernel process to be a set of distributions over positive definite matrices of different sizes, that are consistent under marginalisation (Dawid, 1981; Shah et al., 2014). The two most common kernel processes are the Wishart process and inverse Wishart process, which we write in a slightly unusual form to ensure their expectation is $\mathbf{K}$. We take $\mathbf{G}$ and $\mathbf{G}'$ to be finite dimensional marginals of the underlying Wishart and inverse Wishart process,

$$\mathbf{G} \sim \mathcal{W}\left(\mathbf{K}/N, N\right), \qquad \mathbf{G}' \sim \mathcal{W}^{-1}\left(\delta\mathbf{K}, \delta + (P + 1)\right), \qquad (3a)$$

$$\mathbf{G}^* \sim \mathcal{W}\left(\mathbf{K}^*/N, N\right), \qquad \mathbf{G}'^* \sim \mathcal{W}^{-1}\left(\delta\mathbf{K}^*, \delta + (P^* + 1)\right), \qquad (3b)$$

and where we explicitly give the consistent marginal distributions over $\mathbf{K}^*$, $\mathbf{G}^*$ and $\mathbf{G}'^*$ which are $P^* \times P^*$ principal submatrices of the $P \times P$ matrices $\mathbf{K}$, $\mathbf{G}$ and $\mathbf{G}'$ dropping the same rows and columns. In the inverse-Wishart distribution, $\delta$ is a positive parameter that can be understood as controlling the degree of variability, with larger values for $\delta$ implying smaller variability in $\mathbf{G}'$.

We define a deep kernel process by analogy with a DGP, as a composition of kernel processes, and show in App. A that under sensible assumptions any such composition is itself a kernel process. [1]

### 3.1 DGPs WITH ISOTROPIC KERNELS ARE DEEP WISHART PROCESSES

We consider deep GPs of the form (Fig. 1 top) with $\mathbf{X} \in \mathbb{R}^{P \times N_0}$

$$\mathbf{K}_\ell = \begin{cases} \frac{1}{N_0}\mathbf{X}\mathbf{X}^T & \text{for } \ell = 1, \\ \mathbf{K}\left(\mathbf{G}_{\ell-1}\right) & \text{otherwise,} \end{cases} \qquad (4a)$$

$$\mathrm{P}\left(\mathbf{F}_\ell|\mathbf{K}_\ell\right) = \prod_{\lambda=1}^{N_\ell}\mathcal{N}\left(\mathbf{f}_\lambda^\ell; \mathbf{0}, \mathbf{K}_\ell\right), \qquad (4b)$$

$$\mathbf{G}_\ell = \frac{1}{N_\ell}\mathbf{F}_\ell\mathbf{F}_\ell^T. \qquad (4c)$$

Here, $\mathbf{F}_\ell \in \mathbb{R}^{P \times N_\ell}$ are the $N_\ell$ hidden features in layer $\ell$; $\lambda$ indexes hidden features so $\mathbf{f}_\lambda^\ell$ is a single column of $\mathbf{F}_\ell$, representing the value of the $\lambda$th feature for all training inputs. Note that $\mathbf{K}(\cdot)$ is a

---

[1]Note that we leave the question of the full Kolmogorov extension theorem (Kolmogorov, 1933) for matrices to future work: for our purposes, it is sufficient to work with very large but ultimately finite input spaces as in practice, the input vectors are represented by elements of the finite set of 32-bit or 64-bit floating-point numbers (Sterbenz, 1974).

function that takes a Gram matrix and returns a kernel matrix; whereas $\mathbf{K}_\ell$ is a (possibly random) variable representing a kernel matrix. Note, we have restricted ourselves to kernels, that can be written as functions of the Gram matrix, $\mathbf{G}_\ell$, and do not require the full set of activations, $\mathbf{F}_\ell$. As we describe later, this is not too restrictive, as it includes amongst others all isotropic kernels (i.e. those that can be written as a function of the distance between points Williams & Rasmussen, 2006). Note that we have a number of choices as to how to initialize the kernel in Eq. (4a). The current choice just uses a linear dot-product kernel, rather than immediately applying the kernel function $\mathbf{K}$. This is both to ensure exact equivalence with infinite NNs with bottlenecks (App. C.3) and also to highlight an interesting interpretation of this layer as Bayesian inference over generalised lengthscale hyperparameters in the squared-exponential kernel (App. B e.g. Lalchand & Rasmussen, 2020).

For DGP regression, the outputs, $\mathbf{Y}$, are most commonly given by a likelihood that can be written in terms of the output features, $\mathbf{F}_{L+1}$. For instance, for regression, the distribution of the $\lambda$th output feature column could be

$$\mathrm{P}\left(\mathbf{y}_\lambda|\mathbf{F}_{L+1}\right) = \mathcal{N}\left(\mathbf{y}_\lambda; \mathbf{f}_\lambda^{L+1}, \sigma^2\mathbf{I}\right),\tag{5}$$

but our methods can be used with many other forms for the likelihood, including e.g. classification.

The generative process for the Gram matrices, $\mathbf{G}_\ell$, consists of generating samples from a Gaussian distribution (Eq. 4b), and taking their product with themselves transposed (Eq. 4c). This exactly matches the generative process for a Wishart distribution (Eq. 1), so we can write the Gram matrices, $\mathbf{G}_\ell$, directly in terms of the kernel, without needing to sample features (Fig. 1 bottom),

$$\mathrm{P}\left(\mathbf{G}_1|\mathbf{X}\right) = \mathcal{W}\left(\tfrac{1}{N_1}\left(\tfrac{1}{N_0}\mathbf{X}\mathbf{X}^T\right), N_1\right),\tag{6a}$$

$$\mathrm{P}\left(\mathbf{G}_\ell|\mathbf{G}_{\ell-1}\right) = \mathcal{W}\left(\mathbf{K}\left(\mathbf{G}_{\ell-1}\right)/N_\ell, N_\ell\right), \qquad \text{for } \ell \in \{2, \dots L\},\tag{6b}$$

$$\mathrm{P}\left(\mathbf{F}_{L+1}|\mathbf{G}_L\right) = \prod_{\lambda=1}^{N_{L+1}}\mathcal{N}\left(\mathbf{f}_\lambda^{L+1}; \mathbf{0}, \mathbf{K}\left(\mathbf{G}_L\right)\right).\tag{6c}$$

Except at the output, the model is phrased entirely in terms of positive-definite kernels and Gram matrices, and is consistent under marginalisation (assuming a valid kernel) and is thus a DKP. At a high level, the model can be understood as alternatively sampling a Gram matrix (introducing flexibility in the representation), and nonlinearly transforming the Gram matrix using a kernel (Fig. 2).

This highlights a particularly simple interpretation of the DKP as an autoregressive process. In a standard autoregressive process, we might propagate the current vector, $\mathbf{x}_t$, through a deterministic function, $\mathbf{f}(\mathbf{x}_t)$, and add zero-mean Gaussian noise, $\boldsymbol{\xi}$,

$$\mathbf{x}_{t+1} = \mathbf{f}\left(\mathbf{x}_t\right) + \sigma^2\boldsymbol{\xi} \qquad \text{such that} \qquad \mathbb{E}\left[\mathbf{x}_{t+1}|\mathbf{x}_t\right] = \mathbf{f}\left(\mathbf{x}_t\right).\tag{7}$$

By analogy, the next Gram matrix has expectation centered on a deterministic transformation of the previous Gram matrix,

$$\mathbb{E}\left[\mathbf{G}_\ell|\mathbf{G}_{\ell-1}\right] = \mathbf{K}\left(\mathbf{G}_{\ell-1}\right),\tag{8}$$

so $\mathbf{G}_\ell$ can be written as this expectation plus a zero-mean random variable, $\boldsymbol{\Xi}_\ell$, that can be interpreted as noise,

$$\mathbf{G}_\ell = \mathbf{K}\left(\mathbf{G}_{\ell-1}\right) + \boldsymbol{\Xi}_\ell.\tag{9}$$

Note that $\boldsymbol{\Xi}_\ell$ is not in general positive definite, and may not have an analytically tractable distribution. This noise decreases as $N_\ell$ increases,

$$\mathbb{V}\left[G_{ij}^\ell\right] = \mathbb{V}\left[\Xi_{ij}^\ell\right] = \tfrac{1}{N_\ell}\left(K_{ij}^2(\mathbf{G}_{\ell-1}) + K_{ii}^2(\mathbf{G}_{\ell-1})K_{jj}^2(\mathbf{G}_{\ell-1})\right).\tag{10}$$

Notably, as $N_\ell$ tends to infinity, the Wishart samples converge on their expectation, and the noise disappears, leaving us with a series of deterministic transformations of the Gram matrix. Therefore, we can understand a deep kernel process as alternatively adding "noise" to the kernel by sampling e.g. a Wishart or inverse Wishart distribution ($\mathbf{G}_2$ and $\mathbf{G}_3$ in Fig. 2) and computing a nonlinear transformation of the kernel ($\mathbf{K}(\mathbf{G}_2)$ and $\mathbf{K}(\mathbf{G}_3)$ in Fig. 2)

Remember that we are restricted to kernels that can be written as a function of the Gram matrix,

$$\mathbf{K}_\ell = \mathbf{K}\left(\mathbf{G}_\ell\right) = \mathbf{K}_{\text{features}}\left(\mathbf{F}_\ell\right), \qquad\qquad K_{ij}^\ell = k\left(\mathbf{F}_{i,:}^\ell, \mathbf{F}_{j,:}^\ell\right).\tag{11}$$

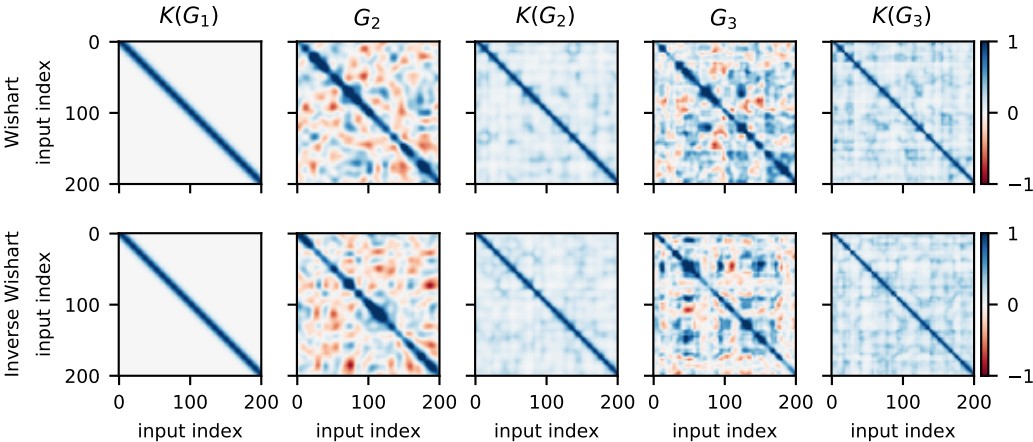

Figure 2: Visualisations of a single prior sample of the kernels and Gram matrices as they pass through the network. We use 1D, equally spaced inputs with a squared exponential kernel. As we transition $\mathbf{K}(\mathbf{G}_{\ell-1}) \to \mathbf{G}_\ell$, we add "noise" by sampling from a Wishart (top) or an inverse Wishart (bottom). As we transition from $\mathbf{G}_\ell$ to $\mathbf{K}(\mathbf{G}_\ell)$, we deterministically transform the Gram matrix using a squared-exponential kernel.

where $\mathbf{K}_{\text{features}}(\cdot)$ takes a matrix of features, $\mathbf{F}_\ell$, and returns the kernel matrix, $\mathbf{K}_\ell$, and $k$ is the usual kernel, which takes two feature vectors (rows of $\mathbf{F}_\ell$) and returns an element of the kernel matrix. This does not include all possible kernels because it is not possible to recover the features from the Gram matrix. In particular, the Gram matrix is invariant to unitary transformations of the features: the Gram matrix is the same for $\mathbf{F}_\ell$ and $\mathbf{F}'_\ell = \mathbf{U}\mathbf{F}_\ell$ where $\mathbf{U}$ is a unitary matrix, such that $\mathbf{U}\mathbf{U}^T = \mathbf{I}$,

$$\mathbf{G}_\ell = \tfrac{1}{N_\ell}\mathbf{F}_\ell\mathbf{F}_\ell^T = \tfrac{1}{N_\ell}\mathbf{F}_\ell\mathbf{U}_\ell\mathbf{U}_\ell^T\mathbf{F}_\ell^T = \tfrac{1}{N_\ell}\mathbf{F}'_\ell\mathbf{F}'^T_\ell. \tag{12}$$

Superficially, this might seem very limiting — leaving us only with dot-product kernels (Williams & Rasmussen, 2006) such as,

$$k(\mathbf{f}, \mathbf{f}') = \mathbf{f} \cdot \mathbf{f}' + \sigma^2. \tag{13}$$

However, in reality, a far broader range of kernels fit within this class. Importantly, isotropic or radial basis function kernels including the squared exponential and Matern depend only on the squared distance between points, $R$, (Williams & Rasmussen, 2006)

$$k(\mathbf{f}, \mathbf{f}') = k(R), \qquad\qquad R = |\mathbf{f} - \mathbf{f}'|^2. \tag{14}$$

These kernels can be written as a function of $\mathbf{G}$, because the matrix of squared distances, $\mathbf{R}$, can be computed from $\mathbf{G}$,

$$R^\ell_{ij} = \tfrac{1}{N_\ell}\sum_{\lambda=1}^{N_\ell}\left(F^\ell_{i\lambda} - F^\ell_{j\lambda}\right)^2 = \tfrac{1}{N_\ell}\sum_{\lambda=1}^{N_\ell}\left(\left(F^\ell_{i\lambda}\right)^2 - 2F^\ell_{i\lambda}F^\ell_{j\lambda} + \left(F^\ell_{j\lambda}\right)^2\right) = G^\ell_{ii} - 2G^\ell_{ij} + G^\ell_{jj}. \tag{15}$$

## 4 VARIATIONAL INFERENCE IN DEEP KERNEL PROCESSES

A key part of the motivation for developing deep kernel processes was that the posteriors over weights in a BNN or over features in a deep GP are extremely complex and multimodal, with a large number of symmetries that are not captured by standard approximate posteriors (MacKay, 1992; Moore, 2016; Pourzanjani et al., 2017). For instance, in the Appendix we show that there are permutation symmetries in the prior and posteriors over weights in BNNs (App. D.1) and rotational symmetries in the prior and posterior over features in deep GPs with isotropic kernels (App. D.2). The inability to capture these symmetries in standard variational posteriors may introduce biases in the parameters inferred by variational inference, because the variational bound is not uniformly tight across the state-space (Turner & Sahani, 2011). Intuitively, these symmetries arise in DGPs with isotropic kernels because the features at the next layer depend only on the kernel matrix at

the previous layer, and this kernel is invariant to unitary transformations of the features (Eq. 12). As such, we can sidestep these complex posterior symmetries by working directly with the Gram matrices as the random variables for variational inference.

We show that DGPs (Sec. 3.1) and infinite NNs with bottlenecks (App. C.3) are deep Wishart processes, so a natural approach would be to define an approximate posterior over the Gram matrices in the deep Wishart process. However, this turns out to be difficult, predominantly because the approximate posterior we would like to use, the non-central Wishart (App. E), has a probability density function that is prohibitively costly and complex to evaluate in the inner loop of a deep learning model (Koev & Edelman, 2006). Instead, we consider an inverse Wishart process prior, for which the inverse Wishart itself makes a good choice of approximate posterior.

### 4.1 THE DEEP INVERSE WISHART PROCESSES

By analogy with Eq. (6), our deep inverse Wishart processes (DIWPs) are given by

$$\mathrm{P}\left(\mathbf{\Omega}\right) = \mathcal{W}^{-1}\left(\delta_1 \mathbf{I}, \delta_1 + N_0 + 1\right), \qquad \text{(with } \mathbf{G}_1 = \tfrac{1}{N_0}\mathbf{X}\mathbf{\Omega}\mathbf{X}^T\text{),} \qquad (16a)$$

$$\mathrm{P}\left(\mathbf{G}_\ell | \mathbf{G}_{\ell-1}\right) = \mathcal{W}^{-1}\left(\mathbf{G}_\ell; \delta_\ell \mathbf{K}\left(\mathbf{G}_{\ell-1}\right), P + 1 + \delta_\ell\right), \qquad \text{for } \ell \in \{2, \dots L\}, \qquad (16b)$$

$$\mathrm{P}\left(\mathbf{F}_{L+1} | \mathbf{G}_L\right) = \prod_{\lambda=1}^{N_{L+1}} \mathcal{N}\left(\mathbf{f}_\lambda^{L+1}; \mathbf{0}, \mathbf{K}\left(\mathbf{G}_L\right)\right), \qquad (16c)$$

remember that $\mathbf{X} \in \mathbb{R}^{P \times N_0}$, $\mathbf{G}_\ell \in \mathbb{R}^{P \times P}$ and $\mathbf{F}_\ell \in \mathbb{R}^{P \times N_{L+1}}$. Note that at the input layer, $\mathbf{K}_0 = \tfrac{1}{N_0}\mathbf{X}\mathbf{X}^T$ may be singular if there are more datapoints than features. Instead of attempting to use a singular Wishart distributions over $\mathbf{G}_1$, which would be complex and difficult to work with (Bodnar & Okhrin, 2008; Bodnar et al., 2016), we instead define an approximate posterior over the full-rank $N_0 \times N_0$ matrix, $\mathbf{\Omega}$, and use $\mathbf{G}_1 = \tfrac{1}{N_0}\mathbf{X}\mathbf{\Omega}\mathbf{X}^T \in \mathbb{R}^{P \times P}$.

Critically, the distributions in Eq. (16b) are consistent under marginalisation as long as $\delta_\ell$ is held constant (Dawid, 1981), with $P$ taken to be the number of input points, or equivalently the size of $\mathbf{K}_{\ell-1}$. Further, the deep inverse Wishart process retains the interpretation as a deterministic transformation of the kernel plus noise because the expectation is,

$$\mathbb{E}\left[\mathbf{G}_\ell | \mathbf{G}_{\ell-1}\right] = \frac{\delta_\ell \mathbf{K}\left(\mathbf{G}_{\ell-1}\right)}{(P + 1 + \delta_\ell) - (P + 1)} = \mathbf{K}\left(\mathbf{G}_{\ell-1}\right). \qquad (17)$$

The resulting inverse Wishart process does not have a direct interpretation as e.g. a deep GP, but does have more appealing properties for variational inference, as it is always full-rank and allows independent control over the approximate posterior mean and variance. Finally, it is important to note that Wishart and inverse Wishart distributions do not differ as much as one might expect; the standard Wishart and standard inverse Wishart distributions have isotropic distributions over the eigenvectors so they only differ in terms of their distributions over eigenvalues, and these are often quite similar, especially if we consider a Wishart model with ResNet-like structure (App. H).

### 4.2 AN APPROXIMATE POSTERIOR FOR THE DEEP INVERSE WISHART PROCESS

Choosing an appropriate and effective form for variational approximate posteriors is usually a difficult research problem. Here, we take inspiration from Ober & Aitchison (2020) by exploiting the fact that the inverse-Wishart distribution is the conjugate prior for the covariance matrix of a multivariate Gaussian. In particular, if we consider an inverse-Wishart prior over $\mathbf{\Sigma} \in \mathbb{R}^{P \times P}$ with mean $\delta \mathbf{\Sigma}_0$, which forms the covariance of Gaussian-distributed matrix, $\mathbf{V} \in \mathbb{R}^{P \times P}$, consisting of columns $\mathbf{v}_\lambda$,

$$\mathrm{P}\left(\mathbf{\Sigma}\right) = \mathcal{W}^{-1}\left(\mathbf{\Sigma}; \delta \mathbf{\Sigma}_0, P + 1 + \delta\right), \qquad (18a)$$

$$\mathrm{P}\left(\mathbf{V} | \mathbf{\Sigma}\right) = \prod_{\lambda=1}^{N_V} \mathcal{N}\left(\mathbf{v}_\lambda; \mathbf{0}, \mathbf{\Sigma}\right), \qquad (18b)$$

$$\mathrm{P}\left(\mathbf{\Sigma} | \mathbf{V}\right) = \mathcal{W}^{-1}\left(\mathbf{\Sigma}; \delta \mathbf{\Sigma}_0 + \mathbf{V}\mathbf{V}^T, P + 1 + \delta + N_V\right). \qquad (18c)$$

Inspired by this exact posterior that is available in simple models, we choose the approximate posterior in our model to be,

$$\mathrm{Q}\left(\mathbf{\Omega}\right) = \mathcal{W}^{-1}\left(\mathbf{\Omega}; \delta_1 \mathbf{I} + \mathbf{V}_1 \mathbf{V}_1^T, \delta_1 + \gamma_1 + (N_0 + 1)\right), \qquad (19a)$$

$$\mathrm{Q}\left(\mathbf{G}_\ell | \mathbf{G}_{\ell-1}\right) = \mathcal{W}^{-1}\left(\mathbf{G}_\ell; \delta_\ell \mathbf{K}\left(\mathbf{G}_{\ell-1}\right) + \mathbf{V}_\ell \mathbf{V}_\ell^T, \delta_\ell + \gamma_\ell + (P + 1)\right), \qquad (19b)$$

$$\mathrm{Q}\left(\mathbf{F}_{L+1} | \mathbf{G}_L\right) = \prod_{\lambda=1}^{N_{L+1}} \mathcal{N}\left(\mathbf{f}_\lambda^{L+1}; \mathbf{\Sigma}_\lambda \mathbf{\Lambda}_\lambda \mathbf{v}_\lambda, \mathbf{\Sigma}_\lambda\right), \quad \text{where} \quad \mathbf{\Sigma}_\lambda = \left(\mathbf{K}^{-1}\left(\mathbf{G}_L\right) + \mathbf{\Lambda}_\lambda\right)^{-1}, \qquad (19c)$$

and where $\mathbf{V}_1$ is a learned $N_0 \times N_0$ matrix, $\{\mathbf{V}_\ell\}_{\ell=2}^L$ are $P \times P$ learned matrices and $\{\gamma_\ell\}_{\ell=1}^L$ are learned non-negative real numbers. For more details about the input layer, see App. F. At the output layer, we take inspiration from the global inducing approximate posterior for DGPs from Ober & Aitchison (2020), with learned parameters being vectors, $\mathbf{v}_\lambda$, and positive definite matrices, $\mathbf{\Lambda}_\lambda$ (see App. G).

In summary, the prior has parameters $\{\delta_\ell\}_{\ell=1}^L$ (which also appears in the approximate posterior), and the posterior has parameters $\{\mathbf{V}_\ell\}_{\ell=1}^L$ and $\{\gamma_\ell\}_{\ell=1}^L$ for the inverse-Wishart hidden layers, and $\{\mathbf{v}_\lambda\}_{\lambda=1}^{N_{L+1}}$ and $\{\mathbf{\Lambda}_\lambda\}_{\lambda=1}^{N_{L+1}}$ at the output. In all our experiments, we optimize all five parameters $(\{\delta_\ell, \mathbf{V}_\ell, \gamma_\ell\}_{\ell=1}^L)$ and $(\{\mathbf{v}_\lambda, \mathbf{\Lambda}_\lambda\}_{\lambda=1}^{N_{L+1}})$, and in addition, for inducing-point methods, we also optimize a single set of "global" inducing inputs, $\mathbf{X}_i \in \mathbb{R}^{P_i \times N_0}$, which are defined only at the input layer.

### 4.3 DOUBLY STOCHASTIC INDUCING-POINT VARIATIONAL INFERENCE IN DEEP INVERSE WISHART PROCESSES

For efficient inference in high-dimensional problems, we take inspiration from the DGP literature (Salimbeni & Deisenroth, 2017) by considering doubly-stochastic inducing-point deep inverse Wishart processes. We begin by decomposing all variables into inducing and training (or test) points $\mathbf{X}_t \in \mathbb{R}^{P_t \times N_0}$,

$$\mathbf{X} = \begin{pmatrix} \mathbf{X}_i \\ \mathbf{X}_t \end{pmatrix}, \qquad \mathbf{F}_{L+1} = \begin{pmatrix} \mathbf{F}_i^{L+1} \\ \mathbf{F}_t^{L+1} \end{pmatrix}, \qquad \mathbf{G}_\ell = \begin{pmatrix} \mathbf{G}_{ii}^\ell & \mathbf{G}_{it}^\ell \\ \mathbf{G}_{ti}^\ell & \mathbf{G}_{tt}^\ell \end{pmatrix}, \qquad (20)$$

where e.g. $\mathbf{G}_{ii}^\ell$ is $P_i \times P_i$ and $\mathbf{G}_{it}^\ell$ is $P_i \times P_t$ where $P_i$ is the number of inducing points, and $P_t$ is the number of testing/training points. Note that $\mathbf{\Omega}$ does not decompose as it is $N_0 \times N_0$. The full ELBO including latent variables for all the inducing and training points is,

$$\mathcal{L} = \mathbb{E}\left[\log \mathrm{P}\left(\mathbf{Y}|\mathbf{F}_{L+1}\right) + \log \frac{\mathrm{P}\left(\mathbf{\Omega}, \{\mathbf{G}_\ell\}_{\ell=2}^L, \mathbf{F}_{L+1}|\mathbf{X}\right)}{\mathrm{Q}\left(\mathbf{\Omega}, \{\mathbf{G}_\ell\}_{\ell=2}^L, \mathbf{F}_{L+1}|\mathbf{X}\right)}\right], \qquad (21)$$

where the expectation is taken over $\mathrm{Q}\left(\mathbf{\Omega}, \{\mathbf{G}_\ell\}_{\ell=2}^L, \mathbf{F}_{L+1}|\mathbf{X}\right)$. The prior is given by combining all terms in Eq. (16) for both inducing and test/train inputs,

$$\mathrm{P}\left(\mathbf{\Omega}, \{\mathbf{G}_\ell\}_{\ell=2}^L, \mathbf{F}_{L+1}|\mathbf{X}\right) = \mathrm{P}\left(\mathbf{\Omega}\right)\left[\prod_{\ell=2}^L \mathrm{P}\left(\mathbf{G}_\ell|\mathbf{G}_{\ell-1}\right)\right]\mathrm{P}\left(\mathbf{F}_{L+1}|\mathbf{G}_L\right), \qquad (22)$$

where the $\mathbf{X}$-dependence enters on the right because $\mathbf{G}_1 = \frac{1}{N_0}\mathbf{X}\mathbf{\Omega}\mathbf{X}^T$. Taking inspiration from Salimbeni & Deisenroth (2017), the full approximate posterior is the product of an approximate posterior over inducing points and the conditional prior for train/test points,

$$\mathrm{Q}\left(\mathbf{\Omega}, \{\mathbf{G}_\ell\}_{\ell=2}^L, \mathbf{F}_{L+1}|\mathbf{X}\right) =$$
$$\mathrm{Q}\left(\mathbf{\Omega}, \{\mathbf{G}_{ii}^\ell\}_{\ell=2}^L, \mathbf{F}_i^{L+1}|\mathbf{X}_i\right)\mathrm{P}\left(\{\mathbf{G}_{it}^\ell\}_{\ell=2}^L, \{\mathbf{G}_{tt}^\ell\}_{\ell=2}^L, \mathbf{F}_t^{L+1}|\mathbf{\Omega}, \{\mathbf{G}_{ii}^\ell\}_{\ell=2}^L, \mathbf{F}_i^{L+1}, \mathbf{X}\right). \qquad (23)$$

And the prior can be written in the same form,

$$\mathrm{P}\left(\mathbf{\Omega}, \{\mathbf{G}_\ell\}_{\ell=2}^L, \mathbf{F}_{L+1}|\mathbf{X}\right) =$$
$$\mathrm{P}\left(\mathbf{\Omega}, \{\mathbf{G}_{ii}^\ell\}_{\ell=2}^L, \mathbf{F}_i^{L+1}|\mathbf{X}_i\right)\mathrm{P}\left(\{\mathbf{G}_{it}^\ell\}_{\ell=2}^L, \{\mathbf{G}_{tt}^\ell\}_{\ell=2}^L, \mathbf{F}_t^{L+1}|\mathbf{\Omega}, \{\mathbf{G}_{ii}^\ell\}_{\ell=2}^L, \mathbf{F}_i^{L+1}, \mathbf{X}\right). \qquad (24)$$

We discuss the second terms (the conditional prior) in Eq. (28). The first terms (the prior and approximate posteriors over inducing points), are given by combining terms in Eq. (16) and Eq. (19),

$$\mathrm{P}\left(\mathbf{\Omega}, \{\mathbf{G}_{ii}^\ell\}_{\ell=2}^L, \mathbf{F}_i^{L+1}|\mathbf{X}_i\right) = \mathrm{P}\left(\mathbf{\Omega}\right)\left[\prod_{\ell=2}^L \mathrm{P}\left(\mathbf{G}_{ii}^\ell|\mathbf{G}_{ii}^{\ell-1}\right)\right]\mathrm{P}\left(\mathbf{F}_i^{L+1}|\mathbf{G}_{ii}^L\right), \qquad (25)$$

$$\mathrm{Q}\left(\mathbf{\Omega}, \{\mathbf{G}_{ii}^\ell\}_{\ell=2}^L, \mathbf{F}_i^{L+1}|\mathbf{X}_i\right) = \mathrm{Q}\left(\mathbf{\Omega}\right)\left[\prod_{\ell=2}^L \mathrm{Q}\left(\mathbf{G}_{ii}^\ell|\mathbf{G}_{ii}^{\ell-1}\right)\right]\mathrm{Q}\left(\mathbf{F}_i^{L+1}|\mathbf{G}_{ii}^L\right). \qquad (26)$$

Substituting Eqs. (23–26) into the ELBO (Eq. 21), the conditional prior cancels and we obtain,

$$\mathcal{L} = \mathbb{E}\left[\log \mathrm{P}\left(\mathbf{Y}|\mathbf{F}_t^{L+1}\right) + \log \frac{\mathrm{Q}\left(\mathbf{\Omega}\right)\left[\prod_{\ell=2}^L \mathrm{Q}\left(\mathbf{G}_{ii}^\ell|\mathbf{G}_{ii}^{\ell-1}\right)\right]\mathrm{Q}\left(\mathbf{F}_i^{L+1}|\mathbf{G}_{ii}^L\right)}{\mathrm{P}\left(\mathbf{\Omega}\right)\left[\prod_{\ell=2}^L \mathrm{P}\left(\mathbf{G}_{ii}^\ell|\mathbf{G}_{ii}^{\ell-1}\right)\right]\mathrm{P}\left(\mathbf{F}_i^{L+1}|\mathbf{G}_{ii}^L\right)}\right]. \qquad (27)$$

Importantly, the first term is a summation across test/train datapoints, and the second term depends only on the inducing points, so as in Salimbeni & Deisenroth (2017) we can compute unbiased estimates of the expectation by taking only a minibatch of datapoints, and we never need to compute the density of the conditional prior in Eq. (28), we only need to be able to sample it.

Finally, to sample the test/training points, conditioned on the inducing points, we need to sample,

$$
\mathrm{P}\left(\{\mathbf{G}_{\mathrm{it}}^{\ell}\}_{\ell=2}^{L}, \{\mathbf{G}_{\mathrm{tt}}^{\ell}\}_{\ell=2}^{L}, \mathbf{F}_{\mathrm{t}}^{L+1} | \mathbf{\Omega}, \{\mathbf{G}_{\mathrm{ii}}^{\ell}\}_{\ell=2}^{L}, \mathbf{F}_{\mathrm{i}}^{L+1}, \mathbf{X}\right) =
$$
$$
\mathrm{P}\left(\mathbf{F}_{\mathrm{t}}^{L+1} | \mathbf{F}_{\mathrm{i}}^{L+1}, \mathbf{G}_{L}\right) \prod_{\ell=2}^{L} \mathrm{P}\left(\mathbf{G}_{\mathrm{it}}^{\ell}, \mathbf{G}_{\mathrm{tt}}^{\ell} | \mathbf{G}_{\mathrm{ii}}^{\ell}, \mathbf{G}_{\ell-1}\right). \quad (28)
$$

The first distribution, $\mathrm{P}\left(\mathbf{F}_{\mathrm{t}}^{L+1} | \mathbf{F}_{\mathrm{i}}^{L+1}, \mathbf{G}_{L}\right)$, is a multivariate Gaussian, and can be evaluated using methods from the GP literature (Williams & Rasmussen, 2006; Salimbeni & Deisenroth, 2017). The difficulties arise for the inverse Wishart terms, $\mathrm{P}\left(\mathbf{G}_{\mathrm{it}}^{\ell}, \mathbf{G}_{\mathrm{tt}}^{\ell} | \mathbf{G}_{\mathrm{ii}}^{\ell}, \mathbf{G}_{\ell-1}\right)$. To sample this distribution, note that samples from the joint over inducing and train/test locations can be written,

$$
\begin{pmatrix} \mathbf{G}_{\mathrm{ii}}^{\ell} & \mathbf{G}_{\mathrm{it}}^{\ell} \\ \mathbf{G}_{\mathrm{ti}}^{\ell} & \mathbf{G}_{\mathrm{tt}}^{\ell} \end{pmatrix} \sim \mathcal{W}^{-1}\left(\begin{pmatrix} \mathbf{\Psi}_{\mathrm{ii}} & \mathbf{\Psi}_{\mathrm{it}} \\ \mathbf{\Psi}_{\mathrm{ti}} & \mathbf{\Psi}_{\mathrm{tt}} \end{pmatrix}, \delta_{\ell} + P_{\mathrm{i}} + P_{\mathrm{t}} + 1\right), \text{ where } \begin{pmatrix} \mathbf{\Psi}_{\mathrm{ii}} & \mathbf{\Psi}_{\mathrm{it}} \\ \mathbf{\Psi}_{\mathrm{ti}} & \mathbf{\Psi}_{\mathrm{tt}} \end{pmatrix} = \delta_{\ell} \mathbf{K}\left(\mathbf{G}_{\ell-1}\right),
$$
$$(29)$$

and where $P_{\mathrm{i}}$ is the number of inducing inputs, and $P_t$ is the number of train/test inputs. Defining the Schur complements,

$$
\mathbf{G}_{\mathrm{tt \cdot i}}^{\ell} = \mathbf{G}_{\mathrm{tt}}^{\ell} - \mathbf{G}_{\mathrm{ti}}^{\ell}\left(\mathbf{G}_{\mathrm{ii}}^{\ell}\right)^{-1} \mathbf{G}_{\mathrm{it}}^{\ell}, \qquad \mathbf{\Psi}_{\mathrm{tt \cdot i}} = \mathbf{\Psi}_{\mathrm{tt}} - \mathbf{\Psi}_{\mathrm{ti}} \mathbf{\Psi}_{\mathrm{ii}}^{-1} \mathbf{\Psi}_{\mathrm{it}}. \quad (30)
$$

We know that $\mathbf{G}_{\mathrm{tt \cdot i}}^{\ell}$ and $\left(\mathbf{G}_{\mathrm{ii}}^{\ell}\right)^{-1} \mathbf{G}_{\mathrm{it}}^{\ell}$ have distribution, (Eaton, 1983)

$$
\mathbf{G}_{\mathrm{tt \cdot i}}^{\ell} | \qquad \mathbf{G}_{\mathrm{ii}}^{\ell}, \mathbf{G}_{\ell-1} \sim \mathcal{W}^{-1}\left(\mathbf{\Psi}_{\mathrm{tt \cdot i}}, \delta_{\ell} + P_{\mathrm{i}} + P_{\mathrm{t}} + 1\right), \quad (31\mathrm{a})
$$
$$
\left(\mathbf{G}_{\mathrm{ii}}^{\ell}\right)^{-1} \mathbf{G}_{\mathrm{it}}^{\ell} | \mathbf{G}_{\mathrm{tt \cdot i}}^{\ell}, \mathbf{G}_{\mathrm{ii}}^{\ell}, \mathbf{G}_{\ell-1} \sim \mathcal{MN}\left(\mathbf{\Psi}_{\mathrm{ii}}^{-1} \mathbf{\Psi}_{\mathrm{it}}, \mathbf{\Psi}_{\mathrm{ii}}^{-1}, \mathbf{G}_{\mathrm{tt \cdot i}}^{\ell}\right), \quad (31\mathrm{b})
$$

where $\mathcal{MN}$ is the matrix normal. Now, $\mathbf{G}_{\mathrm{it}}^{\ell}$ and $\mathbf{G}_{\mathrm{tt}}^{\ell}$, can be recovered by algebraic manipulation. Finally, because of the doubly stochastic form for the objective, we do not need to sample multiple of jointly consistent samples for test points; instead, (and as in DGPs Salimbeni & Deisenroth, 2017) we can independently sample each test point (App. I), which dramatically reduces computational complexity.

We optimize using standard reparameterised variational inference (Kingma & Welling, 2013; Rezende et al., 2014) (Ober & Aitchison, 2020, for details on how to reparameterise samples from the Wishart, see).

## 5 COMPUTATIONAL COMPLEXITY

As in non-deep GPs, the complexity is $\mathcal{O}(P^3)$ for time and $\mathcal{O}(P^2)$ for space for standard DKPs (the $\mathcal{O}(P^3)$ time dependencies emerge e.g. because of inverses and determinants required for the inverse Wishart distributions). For DSVI, there is a $P_{\mathrm{i}}^3$ time and $P_{\mathrm{i}}^2$ space term for the inducing points, because the computations for inducing points are exactly the same as in the non-DSVI case. As we can treat each test/train point independently (App. I), the complexity for test/training points must scale linearly with $P_{\mathrm{t}}$, and this term has $P_{\mathrm{i}}^2$ time scaling, e.g. due to the matrix products in Eq. (30). Thus, the overall complexity for DSVI is $\mathcal{O}(P_{\mathrm{i}}^3 + P_{\mathrm{i}}^2 P_{\mathrm{t}})$ for time and $\mathcal{O}(P_{\mathrm{i}}^2 + P_{\mathrm{i}} P_{\mathrm{t}})$ for space which is exactly the same as non-deep inducing GPs. Thus, and exactly as in non-deep inducing-GPs, by using a small number of inducing points, we are able to convert a cubic dependence on the number of input points into a linear dependence, which gives considerably better scaling.

Surprisingly, this is substantially better than standard DGPs. In standard DGPs, we allow the approximate posterior covariance for each feature to differ (Salimbeni & Deisenroth, 2017), in which case, we are in essence doing standard inducing-GP inference over $N$ hidden features, which gives complexity of $\mathcal{O}(NP_{\mathrm{i}}^3 + NP_{\mathrm{i}}^2 P_{\mathrm{t}})$ for time and $\mathcal{O}(NP_{\mathrm{i}}^2 + NP_{\mathrm{i}} P_{\mathrm{t}})$ for space (Salimbeni & Deisenroth, 2017). It is possible to improve this complexity by restricting the approximate posterior to have the same covariance for each point (but this restriction can be expected to harms performance).

Table 1: Performance in terms of ELBO and predictive log-likelihood for a three-layer (two hidden layer) DGP, NNGP and DIWP on UCI benchmark tasks. Errors are quoted as two standard errors in the *difference* between that method and the best performing method, as in a paired t-test. This is to account for the shared variability that arises due to the use of different test/train splits in the data (20 splits for all but protein, where 5 splits are used Gal & Ghahramani, 2015) some splits are harder for all models, and some splits are easier. Because we consider these differences, errors for the best measure are implicitly included in errors for other measures, and we cannot provide a comparable error for the best method itself.

| metric | dataset | DGP | NNGP | DIWP |
|--------|---------|-----|------|------|
| ELBO | boston | $-1.30 \pm 0.02$ | $-0.31 \pm 0.01$ | $\mathbf{-0.29}$ |
| | concrete | $-0.68 \pm 0.01$ | $-0.40 \pm 0.00$ | $\mathbf{-0.35}$ |
| | energy | $0.59 \pm 0.01$ | $\mathbf{1.47}$ | $\mathbf{1.47 \pm 0.00}$ |
| | kin8nm | $-0.50 \pm 0.01$ | $-0.40 \pm 0.00$ | $\mathbf{-0.33}$ |
| | naval | $-1.42 \pm 0.16$ | $\mathbf{1.38 \pm 0.22}$ | $\mathbf{1.44}$ |
| | power | $-0.04 \pm 0.00$ | $0.00 \pm 0.00$ | $\mathbf{0.01}$ |
| | protein | $\mathbf{-1.07}$ | $-1.11 \pm 0.00$ | $-1.09 \pm 0.01$ |
| | wine | $-1.39 \pm 0.01$ | $-1.17 \pm 0.00$ | $\mathbf{-1.16}$ |
| | yacht | $-0.19 \pm 0.38$ | $1.62 \pm 0.02$ | $\mathbf{1.66}$ |
| test LL | boston | $-3.44 \pm 0.14$ | $-2.46 \pm 0.02$ | $\mathbf{-2.43}$ |
| | concrete | $-3.20 \pm 0.03$ | $-3.13 \pm 0.02$ | $\mathbf{-3.09}$ |
| | energy | $-0.90 \pm 0.05$ | $\mathbf{-0.71}$ | $\mathbf{-0.71 \pm 0.01}$ |
| | kin8nm | $1.05 \pm 0.01$ | $1.10 \pm 0.00$ | $\mathbf{1.12}$ |
| | naval | $2.80 \pm 0.12$ | $\mathbf{5.74}$ | $\mathbf{5.73 \pm 0.21}$ |
| | power | $-2.85 \pm 0.00$ | $-2.83 \pm 0.00$ | $\mathbf{-2.82}$ |
| | protein | $\mathbf{-2.80}$ | $-2.88 \pm 0.01$ | $-2.87 \pm 0.01$ |
| | wine | $-1.18 \pm 0.03$ | $\mathbf{-0.96 \pm 0.01}$ | $\mathbf{-0.95}$ |
| | yacht | $-2.45 \pm 0.49$ | $-0.77 \pm 0.07$ | $\mathbf{-0.67}$ |

## 6  RESULTS

We began by comparing the performance of our deep inverse Wishart process (DIWP) against infinite Bayesian neural networks (known as the neural network Gaussian process or NNGP) and DGPs. To ensure sensible comparisons against the NNGP, we used a ReLU kernel in all models (Cho & Saul, 2009). For all models, we used three layers (two hidden layers and one output layer), with three applications of the kernel. In each case, we used a learned bias and scale for each input feature, and trained for 8000 gradient steps with the Adam optimizer with 100 inducing points, a learning rate of $10^{-2}$ for the first 4000 steps and $10^{-3}$ for the final 4000 steps. For evaluation, we used 100 samples from the final iteration of gradient descent, and for each training step we used 10 samples in the smaller datasets (boston, concrete, energy, wine, yacht), and 1 sample in the larger datasets.

We found that DIWP usually gives better predictive performance and ELBOs. We expected DIWP to be better than (or the same as) the NNGP as the NNGP was a special case of our DIWP (sending $\delta_\ell \to \infty$ sends the variance of the inverse Wishart to zero, so the model becomes equivalent to the NNGP). We found that the DGP performs poorly in comparison to DIWP and NNGPs, and even to past baselines on all datasets except protein (which is by far the largest). This is because we use a ReLU, rather than a squared exponential kernel, as in (Salimbeni & Deisenroth, 2017), and because we used a plain feedforward architecture for all models. In contrast, Salimbeni & Deisenroth (2017) found that good performance with DGPs on even UCI datasets required a complex architecture involving skip connections. Here, we used simple feedforward architectures, both to ensure a fair comparison to the other models, and to avoid the need for an architecture search. In addition, the inverse Wishart process is implicitly able to learn the network "width", $\delta_\ell$, whereas in the DGPs, the width is fixed to be equal to the number of input features, following standard practice in the literature (e.g. Salimbeni & Deisenroth, 2017).

Next, we considered fully-connected networks for small image classification datasets (MNIST and CIFAR-10). We used the same models as in the previous section, with the omission of learned bias and scaling of the inputs. Note that we do not expect these methods to perform well relative to

Table 2: Performance in terms of ELBO test log-likelihood and test accuracy for fully-connected three-layer (two hidden layer) DGPs, NNGP and DIWP on MNIST and CIFAR-10.

| metric | dataset | DGP | NNGP | DIWP |
|--------|---------|-----|------|------|
| ELBO | MNIST | $-0.301 \pm 0.001$ | $-0.268 \pm 0.001$ | $\mathbf{-0.214 \pm 0.001}$ |
| | CIFAR-10 | $-1.735 \pm 0.002$ | $-1.719 \pm 0.001$ | $\mathbf{-1.659 \pm 0.001}$ |
| test LL | MNIST | $-0.130 \pm 0.001$ | $-0.134 \pm 0.002$ | $\mathbf{-0.122 \pm 0.001}$ |
| | CIFAR-10 | $\mathbf{-1.516 \pm 0.002}$ | $-1.539 \pm 0.002$ | $-1.525 \pm 0.003$ |
| test acc. | MNIST | $96.5 \pm 0.1\%$ | $96.5 \pm 0.0\%$ | $\mathbf{96.9 \pm 0.0\%}$ |
| | CIFAR-10 | $46.8 \pm 0.1\%$ | $47.4 \pm 0.1\%$ | $\mathbf{47.7 \pm 0.2\%}$ |

standard methods (e.g. CNNs) for these datasets, as we are using fully-connected networks with only 100 inducing points (whereas e.g. work in the NNGP literature uses the full $60,000 \times 60,000$ covariance matrix). Nonetheless, as the architectures are carefully matched, it provides another opportunity to compare the performance of DIWPs, NNGPs and DGPs. Again, we found that DIWP usually gave statistically significant but perhaps underwhelming gains in predictive performance (except for CIFAR-10 test-log-likelihood, where DIWP lagged by only $0.01$). Importantly, DIWP gives very large improvements in the ELBO, with gains of $0.09$ against DGPs for MNIST and $0.08$ for CIFAR-10 (App. K). For MNIST, remember that the ELBO must be negative (because both the log-likelihood for classification and the KL-divergence term give negative contributions), so the improvement from $-0.301$ to $-0.214$ represents a dramatic change.

## 7 RELATED WORK

Our first contribution was the observation that DGPs with isotropic kernels can be written as deep Wishart processes as the kernel depends only on the Gram matrix. We then gave similar observations for neural networks (App. C.1), infinite neural networks (App. C.2) and infinite network with bottlenecks (App. C.3, also see Aitchison, 2019). These observations motivated us to consider the deep inverse Wishart process prior, which is a novel combination of two pre-existing elements: nonlinear transformations of the kernel (e.g. Cho & Saul, 2009) and inverse Wishart priors over kernels (e.g. Shah et al., 2014). Deep nonlinear transformations of the kernel have been used in the infinite neural network literature (Lee et al., 2017; Matthews et al., 2018) where they form deterministic, parameter-free kernels that do not have any flexibility in the lower-layers (Aitchison, 2019). Likewise, inverse-Wishart distributions have been suggested as priors over covariance matrices (Shah et al., 2014), but they considered a model without nonlinear transformations of the kernel. Surprisingly, without these nonlinear transformations, the inverse Wishart prior becomes equivalent to simply scaling the covariance with a scalar random variable (App. L; Shah et al., 2014). Further linear (inverse) Wishart processes have been used in the financial domain to model how the volatility of asset prices changes over time (Philipov & Glickman, 2006b;a; Asai & McAleer, 2009; Gourieroux & Sufana, 2010; Wilson & Ghahramani, 2010; Heaukulani & van der Wilk, 2019). Importantly, inference in these dynamical (inverse) Wishart processes is often performed by assuming fixed, integer degrees of freedom, and working with underlying Gaussian distributed features. This approach allows one to leverage standard GP techniques (e.g. Kandemir & Hamprecht, 2015; Heaukulani & van der Wilk, 2019), but it is not possible to optimize the degrees of freedom and the posterior over these features usually has rotational symmetries (App. D.2) that are not captured by standard variational posteriors. In contrast, we give a novel doubly-stochastic variational inducing point inference method that operates purely on Gram matrices and thus avoids needing to capture these symmetries.

## 8 CONCLUSIONS

We proposed deep kernel processes which combine nonlinear transformations of the Gram matrix with sampling from matrix-variate distributions such as the inverse Wishart. We showed that DGPs, BNNs (App. C.1), infinite BNNs (App. C.2) and infinite BNNs with bottlenecks (App. C.3) are all instances of DKPs. We defined a new family of deep inverse Wishart processes, and give a novel doubly-stochastic inducing point variational inference scheme that works purely in the space of Gram matrices. DIWP performed better than NNGPs and DGPs on UCI, MNIST and CIFAR-10 benchmarks.

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

## A DKPs ARE KERNEL PROCESSES

We define a generic DKP to be $\mathcal{K}(\mathbf{K})$, for a random matrix $\mathbf{K} \in \mathbb{R}^{P \times P}$. For instance, we could take,

$$\mathcal{K}(\mathbf{K}) = \mathcal{W}(\mathbf{K}, N) \qquad \text{or} \qquad \mathcal{K}(\mathbf{K}) = \mathcal{W}^{-1}(\mathbf{K}, \delta + (P+1)), \qquad (32)$$

where $N$ is a positive integer and $\delta$ is a positive real number. A deep kernel process, $\mathcal{D}$, is the composition of two (or more) underlying kernel processes, $\mathcal{K}_1$ and $\mathcal{K}_2$,

$$\mathbf{G}_1 \sim \mathcal{K}_1(\mathbf{K}), \qquad \mathbf{G}_2 \sim \mathcal{K}_2(\mathbf{G}_1), \qquad (33a)$$

$$\mathbf{G}_2 \sim \mathcal{D}(\mathbf{K}). \qquad (33b)$$

We define $\mathbf{K}^*$, $\mathbf{G}_1^*$ and $\mathbf{G}_2^*$ as principle submatrices of $\mathbf{K}$, $\mathbf{G}_1$ and $\mathbf{G}_2$ respectively, dropping the same rows and columns. To establish that $\mathcal{D}$ is consistent under marginalisation, we use the consistency under marginalisation of $\mathcal{K}_1$ and $\mathcal{K}_2$

$$\mathbf{G}_1^* \sim \mathcal{K}_1(\mathbf{K}^*), \qquad \mathbf{G}_2^* \sim \mathcal{K}_2(\mathbf{G}_1^*), \qquad (34a)$$

and the definition of the $\mathcal{D}$ as the composition of $\mathcal{K}_1$ and $\mathcal{K}_2$ (Eq. 33)

$$\mathbf{G}_2^* \sim \mathcal{D}(\mathbf{K}^*). \qquad (34b)$$

$\mathcal{D}$ is thus consistent under marginalisation, and hence is a kernel process.

Further, note that we can consider $\mathcal{K}$ to be a deterministic distribution that gives mass to only a single $\mathbf{G}$. In that case, $\mathcal{K}$ can be thought of as a deterministic function which must satisfy a corresponding consistency property,

$$\mathbf{G} = \mathcal{K}(\mathbf{K}), \qquad \mathbf{G}^* = \mathcal{K}(\mathbf{K}^*), \qquad (35)$$

and this is indeed satisfied by all deterministic transformations of kernels considered here. In practical terms, as long as $\mathbf{G}$ is always a valid kernel, it is sufficient for the elements of $G_{i \neq j}$ to depend only on $K_{ij}$, $K_{ii}$ and $K_{jj}$ and for $G_{ii}$ to depend only on $K_{jj}$, which is satisfied by e.g. the squared exponential kernel (Eq. 15) and by the ReLU kernel (Cho & Saul, 2009).

## B THE FIRST LAYER OF OUR DEEP GP AS BAYESIAN INFERENCE OVER A GENERALISED LENGTHSCALE

In our deep GP architecture, we first sample $\mathbf{F}_1 \in \mathbb{R}^{P \times N_1}$ from a Gaussian with covariance $\mathbf{K}_0 = \frac{1}{N_0} \mathbf{X} \mathbf{X}^T$ (Eq. 4a). This might seem odd, as the usual deep GP involves passing the input, $\mathbf{X} \in \mathbb{R}^{P \times N_0}$, directly to the kernel function. However, in the standard deep GP framework, the kernel (e.g. a squared exponential kernel) has lengthscale hyperparameters which can be inferred using Bayesian inference. In particular,

$$k_{\text{param}}\left(\tfrac{1}{\sqrt{N_0}} \mathbf{x}_i, \tfrac{1}{\sqrt{N_0}} \mathbf{x}_j\right) = \exp\left(-\tfrac{1}{2N_0} (\mathbf{x}_i - \mathbf{x}_j) \boldsymbol{\Omega} (\mathbf{x}_i - \mathbf{x}_j)^T\right). \qquad (36)$$

where $k_{\text{param}}$ is a new squared exponential kernel that explicitly includes hyperparmeters $\boldsymbol{\Omega} \in \mathbb{R}^{N_0 \times N_0}$, and where $\mathbf{x}_i$ is the $i$th row of $\mathbf{X}$. Typically, in deep GPs, the parameter, $\boldsymbol{\Omega}$, is diagonal, and the diagonal elements correspond to the inverse square of the lengthscale, $l_i$, (i.e. $\Omega_{ii} = 1/l_i^2$). However, in many cases it may be useful to have a non-diagonal scaling. For instance, we could use,

$$\boldsymbol{\Omega} \sim \mathcal{W}\left(\tfrac{1}{N_1} \mathbf{I}, N_1\right), \qquad (37)$$

which corresponds to,

$$\boldsymbol{\Omega} = \mathbf{W} \mathbf{W}^T, \qquad \text{where} \qquad W_{i\lambda} \sim \mathcal{N}\left(0, \tfrac{1}{N_1}\right), \ \mathbf{W} \in \mathbb{R}^{N_0 \times N_1}. \qquad (38)$$

Under our approach, we sample $\mathbf{F} = \mathbf{F}_1$ from Eq. (4b), so $\mathbf{F}$ can be written as,

$$\mathbf{F} = \mathbf{X} \mathbf{W}, \qquad \mathbf{f}_i = \mathbf{x}_i \mathbf{W}, \qquad (39)$$

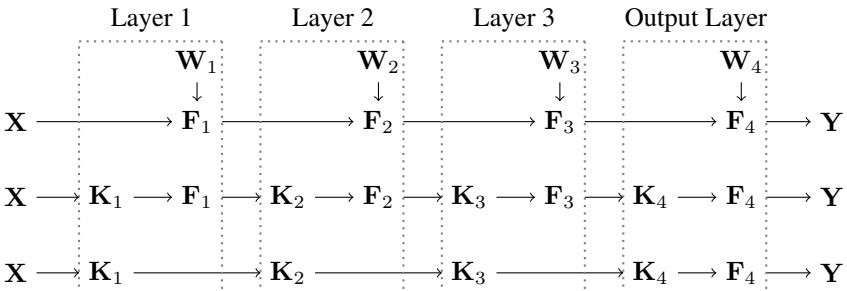

Figure 3: A series of generative models for a standard, finite BNN. **Top**. The standard model, with features, $\mathbf{F}_\ell$, and weights $\mathbf{W}_\ell$ (Eq. 41). **Middle**. Integrating out the weights, the distribution over features becomes Gaussian (Eq. 44), and we explicitly introduce the kernel, $\mathbf{K}_\ell$, as a latent variable. **Bottom**. Integrating out the activations, $\mathbf{F}_\ell$, gives a deep kernel process, albeit one where the distributions $\mathrm{P}\left(\mathbf{K}_\ell|\mathbf{K}_{\ell-1}\right)$ cannot be written down analytically, but where the expectation, $\mathbb{E}\left[\mathbf{K}_\ell|\mathbf{K}_{\ell-1}\right]$ is known (Eq. 45).

where $\mathbf{f}_i$ is the $i$th row of $\mathbf{F}$. Putting this into a squared exponential kernel without a lengthscale parameter,

$$
\begin{aligned}
k(\tfrac{1}{\sqrt{N_0}}\mathbf{f}_i, \tfrac{1}{\sqrt{N_0}}\mathbf{f}_j) &= \exp\left(-\tfrac{1}{2N_0}\left(\mathbf{f}_i - \mathbf{f}_j\right)\left(\mathbf{f}_i - \mathbf{f}_j\right)^T\right), \\
&= \exp\left(-\tfrac{1}{2N_0}\left(\mathbf{x}_i\mathbf{W} - \mathbf{x}_j\mathbf{W}\right)\left(\mathbf{x}_i\mathbf{W} - \mathbf{x}_j\mathbf{W}\right)^T\right), \\
&= \exp\left(-\tfrac{1}{2N_0}\left(\mathbf{x}_i - \mathbf{x}_j\right)\mathbf{W}\mathbf{W}^T\left(\mathbf{x}_i - \mathbf{x}_j\right)^T\right), \\
&= \exp\left(-\tfrac{1}{2N_0}\left(\mathbf{x}_i - \mathbf{x}_j\right)\mathbf{\Omega}\left(\mathbf{x}_i - \mathbf{x}_j\right)^T\right), \\
&= k_{\mathrm{param}}(\tfrac{1}{\sqrt{N_0}}\mathbf{x}_i, \tfrac{1}{\sqrt{N_0}}\mathbf{x}_j).
\end{aligned}
\tag{40}
$$

We find that a parameter-free squared exponential kernel applied to $\mathbf{F}$ is equivalent to a squared-exponential kernel with generalised lengthscale hyperparameters applied to the input.

## C  BNNs as deep kernel processes

Here we show that standard, finite BNNs, infinite BNNs and infinite BNNs with bottlenecks can be understood as deep kernel processes.

### C.1  Stanard finite BNNs (and general DGPs)

Standard, finite BNNs are deep kernel processes, albeit ones which do not admit an analytic expression for the probability density. In particular, the prior for a standard Bayesian neural network (Fig. 3 top) is,

$$
\mathrm{P}\left(\mathbf{W}_\ell\right) = \prod_{\lambda=1}^{N_\ell}\mathcal{N}\left(\mathbf{w}_\lambda^\ell; \mathbf{0}, \mathbf{I}/N_{\ell-1}\right), \qquad \mathbf{W}_\ell \in \mathbb{R}^{N_{\ell-1}\times N_\ell}, \tag{41a}
$$

$$
\mathbf{F}_\ell = \begin{cases} \mathbf{X}\mathbf{W}_1 & \text{for } \ell = 1, \\ \phi\left(\mathbf{F}_{\ell-1}\right)\mathbf{W}_\ell & \text{otherwise}, \end{cases} \qquad \mathbf{F}_\ell \in \mathbb{R}^{P\times N_\ell}, \tag{41b}
$$

where $\mathbf{w}_\lambda^\ell$ is the $\lambda$th column of $\mathbf{W}_\ell$. In the neural-network case, $\phi$ is a pointwise nonlinearity such as a ReLU. Integrating out the weights, the features, $\mathbf{F}_\ell$, become Gaussian distributed, as they depend linearly on the Gaussian distributed weights, $\mathbf{W}_\ell$,

$$
\mathrm{P}\left(\mathbf{F}_\ell|\mathbf{F}_{\ell-1}\right) = \prod_{\lambda=1}^{N_\ell}\mathcal{N}\left(\mathbf{f}_\lambda^\ell; \mathbf{0}, \mathbf{K}_\ell\right) = \mathrm{P}\left(\mathbf{F}_\ell|\mathbf{K}_\ell\right), \tag{42}
$$

where

$$
\mathbf{K}_\ell = \tfrac{1}{N_{\ell-1}}\phi(\mathbf{F}_{\ell-1})\phi^T(\mathbf{F}_{\ell-1}). \tag{43}
$$

Crucially, $\mathbf{F}_\ell$ depends on the previous layer activities, $\mathbf{F}_{\ell-1}$ only through the kernel, $\mathbf{K}_\ell$. As such, we could write a generative model as (Fig. 3 middle),

$$\mathbf{K}_\ell = \begin{cases} \frac{1}{N_0}\mathbf{X}\mathbf{X}^T & \text{for } \ell = 1, \\ \frac{1}{N_{\ell-1}}\phi(\mathbf{F}_{\ell-1})\phi^T(\mathbf{F}_{\ell-1}) & \text{otherwise}, \end{cases} \tag{44a}$$

$$\mathrm{P}\left(\mathbf{F}_\ell | \mathbf{K}_\ell\right) = \prod_{\lambda=1}^{N_\ell} \mathcal{N}\left(\mathbf{f}_\lambda^\ell; \mathbf{0}, \mathbf{K}_\ell\right), \tag{44b}$$

where we have explicitly included the kernel, $\mathbf{K}_\ell$, as a latent variable. This form highlights that BNNs *are* deep GPs, in the sense that $\mathbf{F}_\lambda^\ell$ are Gaussian, with a kernel that depends on the activations from the previous layer. Indeed note that *any* deep GP (i.e. including those with kernels that cannot be written as a function of the Gram matrix) as a kernel, $\mathbf{K}_\ell$, is by definition a matrix that can be written as the outer product of a potentially infinite number of features, $\phi(\mathbf{F}_\ell)$ where we allow $\phi$ to be a much richer class of functions than the usual pointwise nonlinearities (Hofmann et al., 2008). We might now try to follow the approach we took above for deep GPs, and consider a Wishart-distributed Gram matrix, $\mathbf{G}_\ell = \frac{1}{N_\ell}\mathbf{F}_\ell\mathbf{F}_\ell^T$. However, for BNNs we encounter an issue: we are not able to compute the kernel, $\mathbf{K}_\ell$ just using the Gram matrix, $\mathbf{G}_\ell$: we need the full set of features, $\mathbf{F}_\ell$.

Instead, we need an alternative approach to show that a neural network is a deep kernel process. In particular, after integrating out the weights, the resulting distribution is chain-structured (Fig. 3 middle), so in principle we can integrate out $\mathbf{F}_\ell$ to obtain a distribution over $\mathbf{K}_\ell$ conditioned on $\mathbf{K}_{\ell-1}$, giving the DKP model in Fig. 3 (bottom),

$$\mathrm{P}\left(\mathbf{K}_\ell | \mathbf{K}_{\ell-1}\right) = \int d\mathbf{F}_{\ell-1}\, \delta_D\left(\mathbf{K}_\ell - \tfrac{1}{N_\ell}\phi(\mathbf{F}_{\ell-1})\phi^T(\mathbf{F}_{\ell-1})\right) \mathrm{P}\left(\mathbf{F}_{\ell-1} | \mathbf{K}_{\ell-1}\right), \tag{45}$$

where $\mathrm{P}\left(\mathbf{F}_{\ell-1} | \mathbf{K}_{\ell-1}\right)$ is given by Eq. (44b) and $\delta_D$ is the Dirac-delta function, representing the deterministic distribution, $\mathrm{P}\left(\mathbf{K}_\ell | \mathbf{F}_{\ell-1}\right)$ (Eq. 44a). Using this integral to write out the generative process only in terms of $\mathbf{K}_\ell$ gives the deep kernel process in Fig. 3 (bottom). While this distribution exists in principle, it cannot be evaluated analytically. But we can explicitly evaluate the expected value of $\mathbf{K}_\ell$ given $\mathbf{K}_{\ell-1}$ using results from Cho & Saul (2009). In particular, we take Eq. 44a, write out the matrix-multiplication explicitly as a series of vector outer products, and note that as $\mathbf{f}_\lambda^\ell$ is IID across $\ell$, the empirical average is equal to the expectation of a single term, which is computed by Cho & Saul (2009),

$$\mathbb{E}\left[\mathbf{K}_{\ell+1} | \mathbf{K}_\ell\right] = \tfrac{1}{N_\ell}\sum_{\lambda=1}^{N_\ell} \mathbb{E}\left[\phi(\mathbf{f}_\lambda^\ell)\phi^T(\mathbf{f}_\lambda^\ell) | \mathbf{K}_\ell\right] = \mathbb{E}\left[\phi(\mathbf{f}_\lambda^\ell)\phi^T(\mathbf{f}_\lambda^\ell) | \mathbf{K}_\ell\right],$$

$$= \int d\mathbf{f}_\lambda^\ell\, \mathcal{N}\left(\mathbf{f}_\lambda^\ell; \mathbf{0}, \mathbf{K}_\ell\right) \phi(\mathbf{f}_\lambda^\ell)\phi^T(\mathbf{f}_\lambda^\ell) \equiv \mathbf{K}(\mathbf{K}_\ell). \tag{46}$$

Finally, we define this expectation to be $\mathbf{K}(\mathbf{K}_\ell)$ in the case of NNs.

## C.2 INFINITE NNS

We have found that for standard finite neural networks, we were not able to compute the distribution over $\mathbf{K}_\ell$ conditioned on $\mathbf{K}_{\ell-1}$ (Eq. (45)). To resolve this issue, one approach is to consider the limit of an infinitely wide neural network. In this limit, the $\mathbf{K}_\ell$ becomes a deterministic function of $\mathbf{K}_{\ell-1}$, as $\mathbf{K}_\ell$ can be written as the average of $N_\ell$ IID outer products, and as $N_\ell$ grows to infinity, the law of large numbers tells us that the average becomes equal to its expectation,

$$\lim_{N_\ell \to \infty} \mathbf{K}_{\ell+1} = \lim_{N_\ell \to \infty} \tfrac{1}{N_\ell}\sum_{\lambda=1}^{N_\ell}\phi(\mathbf{f}_\lambda^\ell)\phi^T(\mathbf{f}_\lambda^\ell) = \mathbb{E}\left[\phi(\mathbf{f}_\lambda^\ell)\phi^T(\mathbf{f}_\lambda^\ell) | \mathbf{K}_\ell\right] = \mathbf{K}(\mathbf{K}_\ell). \tag{47}$$

## C.3 INFINITE NNS WITH BOTTLENECKS

In infinite NNs, the kernel is deterministic, meaning that there is no flexibility/variability, and hence no capability for representation learning (Aitchison, 2019). Here, we consider infinite networks with bottlenecks that combine the tractability of infinite networks with the flexibility of finite networks (Aitchison, 2019). The trick is to separate flexible, finite linear "bottlenecks" from infinite-width nonlinearities. We keep the nonlinearity infinite in order to ensure that the output kernel is deterministic and can be computed using results from Cho & Saul (2009). In particular, we use finite-width

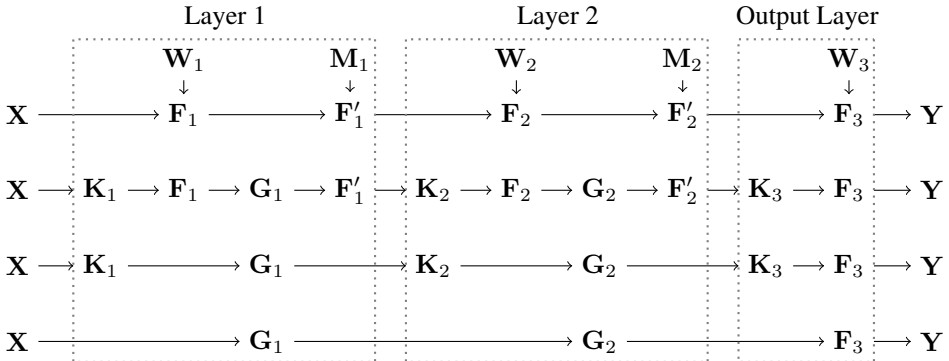

Figure 4: A series of generative models for an infinite network with bottlenecks. **First row**. The standard model. **Second row**. Integrating out the weights. **Third row**. Integrating out the features, the Gram matrices are Wishart-distributed, and the kernels are deterministic. **Last row**. Eliminating all deterministic random variables, we get a model equivalent to that for DGPs (Fig. 1 bottom).

$\mathbf{F}_\ell \in \mathbb{R}^{P \times N_\ell}$ and infinite width $\mathbf{F}'_\ell \in \mathbb{R}^{P \times M_\ell}$, (we send $M_\ell$ to infinity while leaving $N_\ell$ finite),

$$\mathrm{P}\left(\mathbf{W}_\ell\right) = \prod_{\lambda=1}^{N_\ell}\mathcal{N}\left(\mathbf{w}_\lambda^\ell; \mathbf{0}, \mathbf{I}/M_{\ell-1}\right) \quad M_0 = N_0, \tag{48a}$$

$$\mathbf{F}_\ell = \begin{cases} \mathbf{X}\mathbf{W}_\ell & \text{if } \ell = 1, \\ \phi(\mathbf{F}'_{\ell-1})\mathbf{W}_\ell & \text{otherwise,} \end{cases} \tag{48b}$$

$$\mathrm{P}\left(\mathbf{M}_\ell\right) = \prod_{\lambda=1}^{M_\ell}\mathcal{N}\left(\mathbf{m}_\lambda^\ell; \mathbf{0}, \mathbf{I}/N_\ell\right), \tag{48c}$$

$$\mathbf{F}'_\ell = \mathbf{F}_\ell\mathbf{M}_\ell. \tag{48d}$$

This generative process is given graphically in Fig. 4 (top).

Integrating over the expansion weights, $\mathbf{M}_\ell \in \mathbb{R}^{N_\ell \times M_\ell}$, and the bottleneck weights, $\mathbf{W}_\ell \in \mathbb{R}^{M_{\ell-1} \times N_\ell}$, the generative model (Fig. 4 second row) can be rewritten,

$$\mathbf{K}_\ell = \begin{cases} \frac{1}{N_0}\mathbf{X}\mathbf{X}^T & \text{for } \ell = 1, \\ \frac{1}{M_{\ell-1}}\phi\left(\mathbf{F}'_{\ell-1}\right)\phi^T\left(\mathbf{F}'_{\ell-1}\right) & \text{otherwise,} \end{cases} \tag{49a}$$

$$\mathrm{P}\left(\mathbf{F}_\ell|\mathbf{K}_\ell\right) = \prod_{\lambda=1}^{N_\ell}\mathcal{N}\left(\mathbf{f}_\lambda^\ell; \mathbf{0}, \mathbf{K}_\ell\right), \tag{49b}$$

$$\mathbf{G}_\ell = \frac{1}{N_\ell}\mathbf{F}_\ell\mathbf{F}_\ell^T, \tag{49c}$$

$$\mathrm{P}\left(\mathbf{F}'_\ell|\mathbf{G}_\ell\right) = \prod_{\lambda=1}^{M_\ell}\mathcal{N}\left(\mathbf{f}_\lambda'^\ell; \mathbf{0}, \mathbf{G}_\ell\right). \tag{49d}$$

Remembering that $\mathbf{K}_{\ell+1}$ is the empirical mean of $M_\ell$ IID terms, as $M_\ell \to \infty$ it converges on its expectation

$$\lim_{M_\ell \to \infty} \mathbf{K}_{\ell+1} = \lim_{M_\ell \to \infty} \frac{1}{M_\ell}\sum_{\lambda=1}^{N_\ell}\phi\left(\mathbf{f}_\lambda'^\ell\right)\phi^T\left(\mathbf{f}_\lambda'^\ell\right) = \mathbb{E}\left[\phi(\mathbf{f}_\lambda'^\ell)\phi^T(\mathbf{f}_\lambda'^\ell)|\mathbf{G}_\ell\right] = \mathbf{K}(\mathbf{G}_\ell). \tag{50}$$

and we define the limit to be $\mathbf{K}(\mathbf{G}_\ell)$. Note if we use standard (e.g. ReLU) nonlinearities, we can use results from Cho & Saul (2009) to compute $\mathbf{K}(\mathbf{G}_\ell)$. Thus, we get the following generative process,

$$\mathbf{K}_\ell = \begin{cases} \frac{1}{N_0}\mathbf{X}\mathbf{X}^T & \text{for } \ell = 1, \\ \mathbf{K}(\mathbf{G}_{\ell-1}) & \text{otherwise,} \end{cases} \tag{51a}$$

$$\mathrm{P}\left(\mathbf{G}_\ell\right) = \mathcal{W}\left(\mathbf{G}_\ell; \frac{1}{N_\ell}\mathbf{K}_\ell, N_\ell\right). \tag{51b}$$

Finally, eliminating the deterministic kernels, $\mathbf{K}_\ell$, from the model, we obtain exactly the deep GP generative model in Eq. 6 (Fig. C.3 fourth row).

## D   STANDARD APPROXIMATE POSTERIORS OVER FEATURES AND WEIGHTS FAIL TO CAPTURE SYMMETRIES

We have shown that it is possible to represent DGPs and a variety of NNs as deep kernel processes. Here, we argue that standard deep GP approximate posteriors are seriously flawed, and that working with deep kernel processes may alleviate these flaws.

In particular, we show that the true DGP posterior has rotational symmetries and that the true BNN posterior has permutation symmetries that are not captured by standard variational posteriors.

### D.1 PERMUTATION SYMMETRIES IN DNNS POSTERIORS OVER WEIGHTS

Permutation symmetries in neural network posteriors were known in classical work on Bayesian neural networks (e.g. MacKay, 1992). Here, we spell out the argument in full. Taking $\mathbf{P}$ to be a permutation matrix (i.e. a unitary matrix with $\mathbf{P}\mathbf{P}^T = \mathbf{I}$ with one 1 in every row and column), we have,

$$\phi(\mathbf{F})\mathbf{P} = \phi(\mathbf{F}\mathbf{P}). \tag{52}$$

i.e. permuting the input to a nonlinearity is equivalent to permuting its output. Expanding two steps of the recursion defined by Eq. (41b),

$$\mathbf{F}_\ell = \phi(\phi(\mathbf{F}_{\ell-2})\mathbf{W}_{\ell-1})\mathbf{W}_\ell, \tag{53}$$

multiplying by the identity,

$$\mathbf{F}_\ell = \phi(\phi(\mathbf{F}_{\ell-2})\mathbf{W}_{\ell-1})\mathbf{P}\mathbf{P}^T\mathbf{W}_\ell, \tag{54}$$

where $\mathbf{P} \in \mathbb{R}^{N_{\ell-1} \times N_{\ell-1}}$, applying Eq. (52)

$$\mathbf{F}_\ell = \phi(\phi(\mathbf{F}_{\ell-2})\mathbf{W}_{\ell-1}\mathbf{P})\mathbf{P}^T\mathbf{W}_\ell, \tag{55}$$

defining permuted weights,

$$\mathbf{W}'_{\ell-1} = \mathbf{W}_{\ell-1}\mathbf{P}, \qquad\qquad \mathbf{W}'_\ell = \mathbf{P}^T\mathbf{W}_\ell, \tag{56}$$

the output is the same under the original or permuted weights,

$$\mathbf{F}_\ell = \phi(\phi(\mathbf{F}_{\ell-2})\mathbf{W}'_{\ell-1})\mathbf{W}'_\ell = \phi(\phi(\mathbf{F}_{\ell-2})\mathbf{W}_{\ell-1})\mathbf{W}_\ell. \tag{57}$$

Introducing a different perturbation between every pair of layers we get a more general symmetry,

$$\mathbf{W}'_1 = \mathbf{W}_1\mathbf{P}_1, \tag{58a}$$

$$\mathbf{W}_\ell = \mathbf{P}^T_{\ell-1}\mathbf{W}_\ell\mathbf{P}_\ell \quad \text{for } \ell \in \{2, \ldots, L\}, \tag{58b}$$

$$\mathbf{W}'_{L+1} = \mathbf{P}_L\mathbf{W}_{L+1}, \tag{58c}$$

where $\mathbf{P}_\ell \in \mathbb{R}^{N_{\ell-1} \times N_{\ell-1}}$. As the output of the neural network is the same under any of these permutations the likelihoods for original and permuted weights are equal,

$$\mathrm{P}\left(\mathbf{Y}|\mathbf{X}, \mathbf{W}_1, \ldots, \mathbf{W}_{L+1}\right) = \mathrm{P}\left(\mathbf{Y}|\mathbf{X}, \mathbf{W}'_1, \ldots, \mathbf{W}'_{L+1}\right), \tag{59}$$

and as the prior over elements within a weight matrix is IID Gaussian (Eq. 41a), the prior probability density is equal under original and permuted weights,

$$\mathrm{P}\left(\mathbf{W}_1, \ldots, \mathbf{W}_{L+1}\right) = \mathrm{P}\left(\mathbf{W}'_1, \ldots, \mathbf{W}'_{L+1}\right). \tag{60}$$

Thus, the joint probability is invariant to permutations,

$$\mathrm{P}\left(\mathbf{Y}|\mathbf{X}, \mathbf{W}_1, \ldots, \mathbf{W}_{L+1}\right)\mathrm{P}\left(\mathbf{W}_1, \ldots, \mathbf{W}_{L+1}\right) = \mathrm{P}\left(\mathbf{Y}|\mathbf{X}, \mathbf{W}'_1, \ldots, \mathbf{W}'_{L+1}\right)\mathrm{P}\left(\mathbf{W}'_1, \ldots, \mathbf{W}'_{L+1}\right), \tag{61}$$

and applying Bayes theorem, the posterior is invariant to permutations,

$$\mathrm{P}\left(\mathbf{W}_1, \ldots, \mathbf{W}_{L+1}|\mathbf{Y}, \mathbf{X}\right) = \mathrm{P}\left(\mathbf{W}'_1, \ldots, \mathbf{W}'_{L+1}|\mathbf{Y}, \mathbf{X}\right). \tag{62}$$

Due in part to these permutation symmetries, the posterior distribution over weights is extremely complex and multimodal. Importantly, it is not possible to capture these symmetries using standard variational posteriors over weights, such as factorised posteriors, but it is not necessary to capture these symmetries if we work with Gram matrices and kernels, which are invariant to permutations (and other unitary transformations; Eq. 12).

## D.2 ROTATIONAL SYMMETRIES IN DEEP GP POSTERIORS

To show that deep GP posteriors are invariant to unitary transformations, $\mathbf{U}_\ell \in \mathbb{R}^{N_\ell \times N_\ell}$, where $\mathbf{U}_\ell \mathbf{U}_\ell^T = \mathbf{I}$, we define transformed features, $\mathbf{F}'_\ell$,

$$\mathbf{F}'_\ell = \mathbf{F}_\ell \mathbf{U}_\ell. \tag{63}$$

To evaluate $\mathrm{P}\left(\mathbf{F}'_\ell | \mathbf{F}'_{\ell-1}\right)$, we begin by substituting for $\mathbf{F}'_{\ell-1}$,

$$\mathrm{P}\left(\mathbf{F}'_\ell | \mathbf{F}'_{\ell-1}\right) = \prod_{\lambda=1}^{N_\ell} \mathcal{N}\left(\mathbf{f}'^\ell_\lambda; \mathbf{0}, \mathbf{K}\left(\tfrac{1}{N_{\ell-1}} \mathbf{F}'_{\ell-1} \mathbf{F}'^T_{\ell-1}\right)\right), \tag{64}$$

$$= \prod_{\lambda=1}^{N_\ell} \mathcal{N}\left(\mathbf{f}'^\ell_\lambda; \mathbf{0}, \mathbf{K}\left(\tfrac{1}{N_{\ell-1}} \mathbf{F}_{\ell-1} \mathbf{U}_{\ell-1} \mathbf{U}^T_{\ell-1} \mathbf{F}^T_{\ell-1}\right)\right), \tag{65}$$

$$= \prod_{\lambda=1}^{N_\ell} \mathcal{N}\left(\mathbf{f}'^\ell_\lambda; \mathbf{0}, \mathbf{K}\left(\tfrac{1}{N_{\ell-1}} \mathbf{F}_{\ell-1} \mathbf{F}^T_{\ell-1}\right)\right), \tag{66}$$

$$= \mathrm{P}\left(\mathbf{F}'_\ell | \mathbf{F}_{\ell-1}\right). \tag{67}$$

To evaluate $\mathrm{P}\left(\mathbf{F}'_\ell | \mathbf{F}_{\ell-1}\right)$, we substitute for $\mathbf{F}'_\ell$ in the explicit form for the multivariate Gaussian probability density,

$$\mathrm{P}\left(\mathbf{F}'_\ell | \mathbf{F}_{\ell-1}\right) = -\tfrac{1}{2} \operatorname{Tr}\left(\mathbf{F}'^T_\ell \mathbf{K}^{-1}_{\ell-1} \mathbf{F}'_\ell\right) + \text{const}, \tag{68}$$

$$= -\tfrac{1}{2} \operatorname{Tr}\left(\mathbf{K}^{-1}_{\ell-1} \mathbf{F}'_\ell \mathbf{F}'^T_\ell\right) + \text{const}, \tag{69}$$

$$= -\tfrac{1}{2} \operatorname{Tr}\left(\mathbf{K}^{-1}_{\ell-1} \mathbf{F}_\ell \mathbf{U}_\ell \mathbf{U}^T_\ell \mathbf{F}^T_\ell\right) + \text{const}, \tag{70}$$

$$= -\tfrac{1}{2} \operatorname{Tr}\left(\mathbf{K}^{-1}_{\ell-1} \mathbf{F}_\ell \mathbf{F}^T_\ell\right) + \text{const}, \tag{71}$$

$$= \mathrm{P}\left(\mathbf{F}_\ell | \mathbf{F}_{\ell-1}\right). \tag{72}$$

where $\mathbf{K}_{\ell-1} = \mathbf{K}\left(\tfrac{1}{N_{\ell-1}} \mathbf{F}_{\ell-1} \mathbf{F}^T_{\ell-1}\right)$, and the constant depends only on $\mathbf{F}_{\ell-1}$. Combining these derivations, each of these conditionals is invariant to rotations of $\mathbf{F}_\ell$ and $\mathbf{F}_{\ell-1}$,

$$\mathrm{P}\left(\mathbf{F}'_\ell | \mathbf{F}'_{\ell-1}\right) = \mathrm{P}\left(\mathbf{F}'_\ell | \mathbf{F}_{\ell-1}\right) = \mathrm{P}\left(\mathbf{F}_\ell | \mathbf{F}_{\ell-1}\right). \tag{73}$$

The same argument can straightforwardly be extended to the inputs, $\mathrm{P}\left(\mathbf{F}_1 | \mathbf{X}\right)$,

$$\mathrm{P}\left(\mathbf{F}'_1 | \mathbf{X}\right) = \mathrm{P}\left(\mathbf{F}_1 | \mathbf{X}\right), \tag{74}$$

and to the final probability density, for output activations, $\mathbf{F}_{L+1}$ which is not invariant to permutations,

$$\mathrm{P}\left(\mathbf{F}_{L+1} | \mathbf{F}'_L\right) = \mathrm{P}\left(\mathbf{F}_{L+1} | \mathbf{F}'_L\right), \tag{75}$$

Therefore, we have,

$$\mathrm{P}\left(\mathbf{F}'_1, \ldots, \mathbf{F}'_L, \mathbf{F}_{L+1}, \mathbf{Y} | \mathbf{X}\right) = \mathrm{P}\left(\mathbf{Y} | \mathbf{F}_{L+1}\right) \mathrm{P}\left(\mathbf{F}_{L+1} | \mathbf{F}'_L\right) \left(\prod_{\ell=2}^{L} \mathrm{P}\left(\mathbf{F}'_\ell | \mathbf{F}'_{\ell-1}\right)\right) \mathrm{P}\left(\mathbf{F}'_1 | \mathbf{X}\right), \tag{76}$$

$$= \mathrm{P}\left(\mathbf{Y} | \mathbf{F}_{L+1}\right) \mathrm{P}\left(\mathbf{F}_{L+1} | \mathbf{F}_L\right) \left(\prod_{\ell=2}^{L} \mathrm{P}\left(\mathbf{F}_\ell | \mathbf{F}_{\ell-1}\right)\right) \mathrm{P}\left(\mathbf{F}_1 | \mathbf{X}\right), \tag{77}$$

$$= \mathrm{P}\left(\mathbf{F}_1, \ldots, \mathbf{F}_L, \mathbf{F}_{L+1}, \mathbf{Y} | \mathbf{X}\right). \tag{78}$$

Therefore, applying Bayes theorem the posterior is invariant to rotations,

$$\mathrm{P}\left(\mathbf{F}'_1, \ldots, \mathbf{F}'_L, \mathbf{F}_{L+1} | \mathbf{X}, \mathbf{Y}\right) = \mathrm{P}\left(\mathbf{F}_1, \ldots, \mathbf{F}_L, \mathbf{F}_{L+1} | \mathbf{X}, \mathbf{Y}\right). \tag{79}$$

Importantly, these posterior symmetries are not captured by standard variational posteriors with non-zero means (e.g. Salimbeni & Deisenroth, 2017).

## D.3 THE TRUE POSTERIOR OVER FEATURES IN A DGP HAS ZERO MEAN

We can use symmetry to show that the posterior of $\mathbf{F}_\ell$ has zero mean. We begin by writing the expectation as an integral,

$$\mathbb{E}\left[\mathbf{F}_\ell | \mathbf{F}_{\ell-1}, \mathbf{F}_{\ell+1}\right] = \int d\mathbf{F} \; \mathbf{F} \, \mathrm{P}\left(\mathbf{F}_\ell{=}\mathbf{F} | \mathbf{F}_{\ell-1}, \mathbf{F}_{\ell+1}\right). \tag{80}$$

Changing variables in the integral to $\mathbf{F}' = -\mathbf{F}$, and noting that the absolute value of the Jacobian is 1, we have

$$= \int d\mathbf{F}' \; (-\mathbf{F}') \, \mathrm{P} \left( \mathbf{F}_\ell = (-\mathbf{F}') \, | \mathbf{F}_{\ell-1}, \mathbf{F}_{\ell+1} \right), \tag{81}$$

using the symmetry of the posterior,

$$= \int d\mathbf{F}' \; (-\mathbf{F}') \, \mathrm{P} \left( \mathbf{F}_\ell = \mathbf{F}' | \mathbf{F}_{\ell-1}, \mathbf{F}_{\ell+1} \right), \tag{82}$$

$$= - \mathbb{E} \left[ \mathbf{F}_\ell | \mathbf{F}_{\ell-1}, \mathbf{F}_{\ell+1} \right], \tag{83}$$

the expectation is equal to minus itself, so it must be zero

$$\mathbb{E} \left[ \mathbf{F}_\ell | \mathbf{F}_{\ell-1}, \mathbf{F}_{\ell+1} \right] = \mathbf{0}. \tag{84}$$

## E  DIFFICULTIES WITH VI IN DEEP WISHART PROCESSES

The deep Wishart generative process is well-defined as long as we admit nonsingular Wishart distributions (Uhlig, 1994; Srivastava et al., 2003). The issue comes when we try to form a variational approximate posterior over low-rank positive definite matrices. This is typically the case because the number of datapoints, $P$ is usually far larger than the number of features. In particular, the only convenient distribution over low-rank positive semidefinite matrices is the Wishart itself,

$$\mathrm{Q} \left( \mathbf{G}_\ell \right) = \mathcal{W} \left( \mathbf{G}_\ell; \tfrac{1}{N_\ell} \boldsymbol{\Psi}, N_\ell \right). \tag{85}$$

However, a key feature of most variational approximate posteriors is the ability to increase and decrease the variance, independent of other properties such as the mean, and in our case the rank of the matrix. For a Wishart, the mean and variance are given by,

$$\underset{\mathrm{Q}(\mathbf{G}_\ell)}{\mathbb{E}} \left[ \mathbf{G}_\ell \right] = \boldsymbol{\Psi}, \tag{86}$$

$$\underset{\mathrm{Q}(\mathbf{G}_\ell)}{\mathbb{V}} \left[ G_{ij}^\ell \right] = \tfrac{1}{N_\ell} \left( \Psi_{ij}^2 + \Psi_{ii} \Psi_{jj} \right). \tag{87}$$

Initially, this may look fine: we can increase or decrease the variance by changing $N_\ell$. However, remember that $N_\ell$ is the degrees of freedom, which controls the rank of the matrix, $\mathbf{G}_\ell$. As such, $N_\ell$ is fixed by the prior: the prior and approximate posterior must define distributions over matrices of the same rank. And once $N_\ell$ is fixed, we no longer have independent control over the variance.

To go about resolving this issue, we need to find a distribution over low-rank matrices with independent control of the mean and variance. The natural approach is to use a non-central Wishart, defined as the outer product of Gaussian-distributed vectors with non-zero means. While this distribution is easy to sample from and does give independent control over the rank, mean and variance, its probability density is prohibitively costly and complex to evaluate (Koev & Edelman, 2006).

## F  SINGULAR (INVERSE) WISHART PROCESSES AT THE INPUT LAYER

In almost all cases of interest, our the kernel functions $\mathbf{K}(\mathbf{G})$ return full-rank matrices, so we can use standard (inverse) Wishart distributions, which assume that the input matrix is full-rank. However, this is not true at the input layer as $\mathbf{K}_0 = \frac{1}{N_0} \mathbf{X} \mathbf{X}^T$ will often be low-rank. This requires us to use singular (inverse) Wishart distributions which in general are difficult to work with (Uhlig, 1994; Srivastava et al., 2003; Bodnar & Okhrin, 2008; Bodnar et al., 2016). As such, instead we exploit knowledge of the input features to work with a smaller, full-rank matrix, $\boldsymbol{\Omega} \in \mathbb{R}^{N_0 \times N_0}$, where, remember, $N_0$ is the number of input features in $\mathbf{X}$. For a deep Wishart process,

$$\tfrac{1}{N_0} \mathbf{X} \boldsymbol{\Omega} \mathbf{X}^T = \mathbf{G}_1 \sim \mathcal{W} \left( \tfrac{1}{N_1} \mathbf{K}_0, N_1 \right), \qquad \text{where} \quad \boldsymbol{\Omega} \sim \mathcal{W} \left( \tfrac{1}{N_1} \mathbf{I}, N_1 \right), \tag{88}$$

and for a deep inverse Wishart process,

$$\tfrac{1}{N_0} \mathbf{X} \boldsymbol{\Omega} \mathbf{X}^T = \mathbf{G}_1 \sim \mathcal{W}^{-1} \left( \delta_1 \mathbf{K}_0, \delta_1 + P + 1 \right), \quad \text{where} \quad \boldsymbol{\Omega} \sim \mathcal{W}^{-1} \left( \delta_1 \mathbf{I}, \delta_1 + N_0 + 1 \right). \tag{89}$$

Now, we are able to use the full-rank matrix, $\boldsymbol{\Omega}$ rather than the low-rank matrix, $\mathbf{G}_1$ as the random variable for variational inference. For the approximate posterior over $\boldsymbol{\Omega}$, in a deep inverse Wishart process, we use

$$\mathrm{Q} \left( \boldsymbol{\Omega} \right) = \mathcal{W}^{-1} \left( \delta_1 \mathbf{I} + \mathbf{V}_1 \mathbf{V}_1^T, \delta_1 + \gamma_1 + (N_0 + 1) \right). \tag{90}$$

Note in the usual case where there are fewer inducing points than input features, then the matrix $\mathbf{K}_0$ will be full-rank, and we can work with $\mathbf{G}_1$ as the random variable as usual.

## G  APPROXIMATE POSTERIORS OVER OUTPUT FEATURES

To define approximate posteriors over inducing outputs, we are inspired by global inducing point methods (Ober & Aitchison, 2020). In particular, we take the approximate posterior to be the prior, multiplied by a "pseudo-likelihood",

$$\mathrm{Q}\left(\mathbf{F}_{L+1}|\mathbf{G}_L\right) \propto \mathrm{P}\left(\mathbf{F}_{L+1}|\mathbf{G}_L\right) \prod_{\lambda=1}^{N_{L+1}} \mathcal{N}\left(\mathbf{v}_\lambda; \mathbf{f}_\lambda^{L+1}, \mathbf{\Lambda}_\lambda^{-1}\right). \tag{91}$$

This is valid both for global inducing inputs and (for small datasets) training inputs, and the key thing to remember is that in either case, for any given input (e.g. an MNIST handwritten 2), there is a desired output (e.g. the class-label "2"), and the top-layer global inducing outputs, $\mathbf{v}_\lambda$, express these desired outcomes. Substituting for the prior,

$$\mathrm{Q}\left(\mathbf{F}_{L+1}|\mathbf{G}_L\right) \propto \prod_{\lambda=1}^{N_{L+1}} \mathcal{N}\left(\mathbf{f}_\lambda^{L+1}; \mathbf{0}, \mathbf{K}(\mathbf{G}_L)\right) \mathcal{N}\left(\mathbf{v}_\lambda; \mathbf{f}_\lambda^{L+1}, \mathbf{\Lambda}_\lambda^{-1}\right), \tag{92}$$

and computing this value gives the approximate posterior in the main text (Eq. 19).

## H  USING EIGENVALUES TO COMPARE DEEP WISHART, DEEP RESIDUAL WISHART AND INVERSE WISHART PRIORS

One might be concerned that the deep inverse Wishart processes in which we can easily perform inference are different to the deep Wishart processes corresponding to BNNs (Sec. C.1) and infinite NNs with bottlenecks (App. C.3). To address these concerns, we begin by noting that the (inverse) Wishart priors can be written in terms of samples from the standard (inverse) Wishart

$$\mathbf{G} = \mathbf{L}\mathbf{\Omega}\mathbf{L}^T, \qquad\qquad \mathbf{G}' = \mathbf{L}\mathbf{\Omega}'\mathbf{L}^T, \tag{93}$$

where $\mathbf{K} = \mathbf{L}\mathbf{L}^T$ such that,

$$\mathbf{\Omega} \sim \mathcal{W}\left(\tfrac{1}{N}\mathbf{I}, N\right), \qquad\qquad \mathbf{\Omega}' \sim \mathcal{W}^{-1}\left(N\mathbf{I}, \lambda N\right), \tag{94}$$

$$\mathbf{G} \sim \mathcal{W}\left(\tfrac{1}{N}\mathbf{K}, N\right), \qquad\qquad \mathbf{G}' \sim \mathcal{W}^{-1}\left(N\mathbf{K}, \lambda N\right). \tag{95}$$

Note that as the standard Wishart and inverse Wishart have uniform distributions over the eigenvectors (Shah et al., 2014), they differ only in the distribution over eigenvalues of $\mathbf{\Omega}$ and $\mathbf{\Omega}'$. We plotted the eigenvalue histogram for samples from a Wishart distribution with $N = P = 2000$ (Fig. 5 top left). This corresponds to an IID Gaussian prior over weights, with 2000 features in the input and output layers. Notably, there are many very small eigenvalues, which are undesirable as they eliminate information present in the input. To eliminate these very small eigenvalues, a common approach is to use a ResNet-inspired architecture (which is done even in the deep GP literature, e.g. Salimbeni & Deisenroth, 2017). To understand the eigenvalues in a residual layer, we define a Res$\mathcal{W}$ distribution by taking the outer product of a weight matrix with itself,

$$\mathbf{W}\mathbf{W}^T = \mathbf{\Omega}'' \sim \mathrm{Res}\mathcal{W}\left(N, \alpha\right), \tag{96}$$

where the weight matrix is IID Gaussian, plus the identity matrix, with the identity matrix weighted as $\alpha$,

$$\mathbf{W} = \tfrac{1}{\sqrt{1+\alpha^2}}\left(\sqrt{\tfrac{1}{N}}\boldsymbol{\xi} + \alpha\mathbf{I}\right), \qquad\qquad \xi_{i,\lambda} \sim \mathcal{N}\left(0, 1\right). \tag{97}$$

With $\alpha = 1$, there are still many very small eigenvalues, but these disappear as $\alpha$ increases. We compared these distributions to inverse Wishart distributions (Fig. 5 bottom) with varying degrees of freedom. For all degrees of freedom, we found that inverse Wishart distributions do not produce very small eigenvalues, which would eliminate information. As such, these eigenvalue distributions resemble those for Res$\mathcal{W}$ with $\alpha$ larger than 1.

## I  DOUBLY STOCHASTIC VARIATIONAL INFERENCE IN DEEP INVERSE WISHART PROCESSES

Due to the doubly stochastic results in Sec. 4.3, we only need to compute the conditional distribution over a single test/train point (we do not need the joint distribution over a number of test points). As such, we can decompose $\mathbf{G}$ and $\mathbf{Psi}$ as,

$$\mathbf{G}_\ell = \begin{pmatrix} \mathbf{G}_{\mathrm{ii}}^\ell & \mathbf{g}_{\mathrm{it}}^{\ell T} \\ \mathbf{g}_{\mathrm{it}}^\ell & g_{\mathrm{tt}}^\ell \end{pmatrix}, \qquad\qquad \mathbf{\Psi} = \begin{pmatrix} \mathbf{\Psi}_{\mathrm{ii}} & \boldsymbol{\psi}_{\mathrm{it}}^T \\ \boldsymbol{\psi}_{\mathrm{it}} & \psi_{\mathrm{tt}} \end{pmatrix}, \tag{98}$$

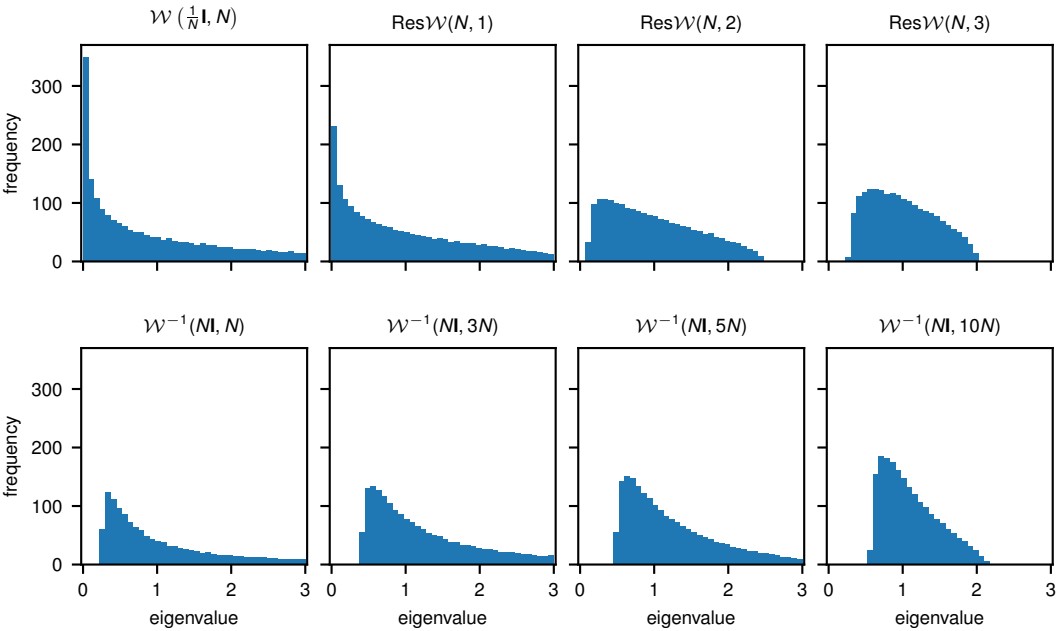

Figure 5: Eigenvalue histograms for a single sample from the labelled distribution, with $N = 2000$.

where $\mathbf{G}_{\text{ii}}^{\ell}$, $\boldsymbol{\Psi}_{\text{ii}} \in \mathbb{R}^{P_i \times P_i}$, $\mathbf{g}_{\text{it}}^{\ell} \in \mathbb{R}^{P_i times 1}$ and $\boldsymbol{\psi}_{\text{it}} \in \mathbb{R}^{P_i \times 1}$ are column-vectors, and $g_{\text{tt}}^{\ell}$ and $\psi_{\text{tt}}$ are scalars. Taking the results in Eq. (31) to the univariate case,

$$g_{\text{tt·i}}^{\ell} = g_{\text{tt}}^{\ell} - \mathbf{g}_{\text{it}}^{T\ell} \left( \mathbf{G}_{\text{ii}}^{\ell} \right)^{-1} \mathbf{g}_{\text{it}}^{\ell}, \qquad\qquad \psi_{\text{tt·i}} = \psi_{\text{tt}} - \boldsymbol{\psi}_{\text{it}}^{T} \boldsymbol{\Psi}_{\text{ii}}^{-1} \boldsymbol{\psi}_{\text{it}}. \tag{99}$$

As $g_{\text{tt·i}}^{\ell}$ is univariate, its distribution becomes Inverse Gamma,

$$g_{\text{tt·i}}^{\ell} | \mathbf{G}_{\text{ii}}^{\ell}, \mathbf{G}_{\ell-1} \sim \text{InverseGamma} \left( \alpha = \tfrac{1}{2} \left( \delta_\ell + P_{\text{t}} + P_{\text{i}} + 1 \right), \beta = \tfrac{1}{2} \psi_{\text{tt·i}} \right). \tag{100}$$

As $\mathbf{g}_{it}^{\ell}$ is a vector rather than a matrix, its distribution becomes Gaussian,

$$\left( \mathbf{G}_{\text{ii}}^{\ell} \right)^{-1} \mathbf{g}_{\text{it}}^{\ell} | g_{\text{tt·i}}^{\ell}, \mathbf{G}_{\text{ii}}^{\ell}, \mathbf{G}_{\ell-1} \sim \mathcal{N} \left( \boldsymbol{\Psi}_{\text{ii}}^{-1} \boldsymbol{\psi}_{\text{it}}, g_{\text{tt·i}}^{\ell} \boldsymbol{\Psi}_{\text{ii}}^{-1} \right). \tag{101}$$

## J  SAMPLES FROM THE 1D PRIOR AND APPROXIMATE POSTERIOR

First, we drew samples from a one-layer (top) and two-layer (bottom) deep inverse Wishart process, with a squared-exponential kernel (Fig. 6). We found considerable differences in the function family corresponding to different prior samples of the top-layer Gram matrix, $\mathbf{G}_L$ (panels). While differences across function classes in a one-layer IW process can be understood as equivalent to doing inference over a prior on the lengthscale, this is not true of the two-layer process, and to emphasise this, the panels for two-layer samples all have the same first layer sample (equivalent to choosing a lengthscale), but different samples from the Gram matrix at the second layer. The two-layer deep IW process panels use the same, fixed input layer, so variability in the function class arises only from sampling $\mathbf{G}_2$.

Next, we exploited kernel flexibilities in IW processes by training a one-layer deep IW model with a fixed kernel bandwidth on data generated from various bandwidths. The first row in Figure 7 shows posterior samples from one-layer deep IW processes trained on different datasets. For each panel, we first sampled five full $\mathbf{G}_1$ matrices using Eq.(31a) and (31b). Then for each $\mathbf{G}_1$, we use Gaussian conditioning to get a posterior distribution on testing locations and drew one sample from the posterior plotted as a single line. Remarkably, these posterior samples exhibited wiggling behaviours that were consistent with training data even outside the training range, which highlighted the additional kernel flexibility in IW processes. On the other hand, when model bandwidth was fixed, samples from vanilla GPs with fixed bandwidth in the second row displayed almost identical shapes outside the training range across different sets of training data.

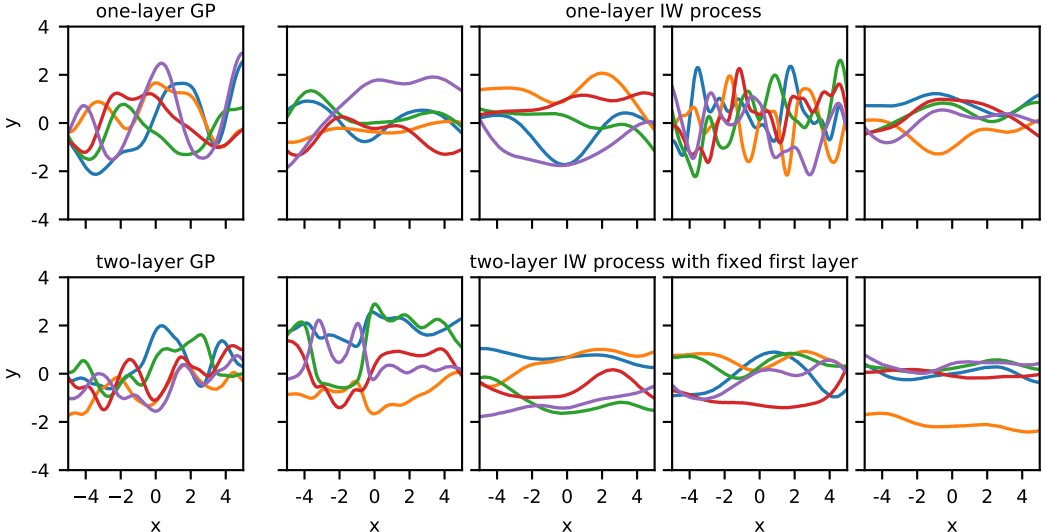

Figure 6: Samples from a one-layer (top) and a two-layer (bottom) deep IW process prior (Eq. 16). On the far left, we have included a set of samples from a GP with the same kernel, for comparison. This GP is equivalent to sending $\delta_0 \to \infty$ in the one-layer deep IW process and additionally sending $\delta_1 \to \infty$ in the two-layer deep IW process. All of the deep IW process panels use the same squared-exponential kernel with bandwidth 1. and $\delta_0 = \delta_1 = 0$. For each panel, we draw a single sample of the top-layer Gram matrix, $\mathbf{G}_L$, then draw multiple GP-distributed functions, conditioned on that Gram matrix.

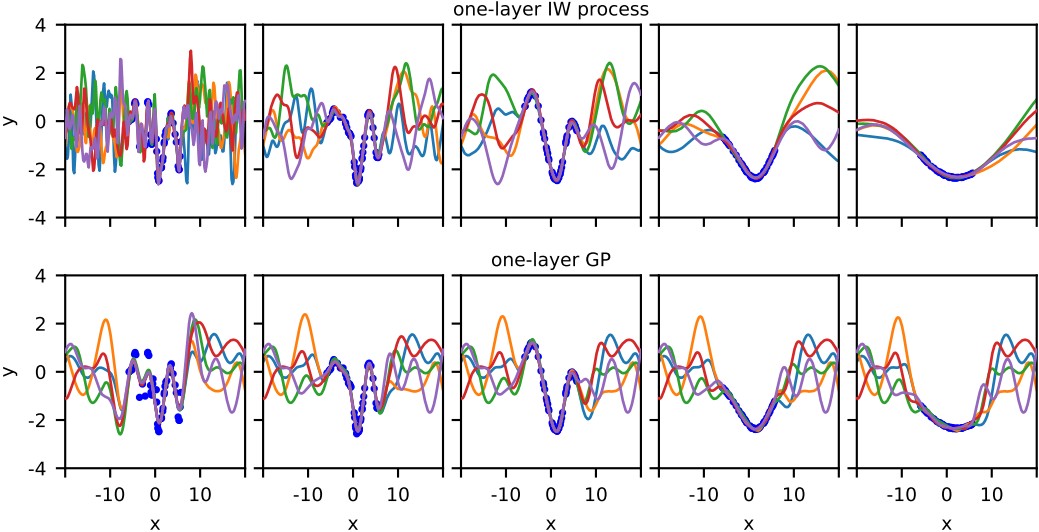

Figure 7: The additional flexibility in a one-layer deep IW process can be used to capture mismatch in the kernel. We plot five posterior function samples from trained IW processes in the first row, and samples from trained GPs below. We generate different sets of training data from a GP with different kernel bandwidths $(0.5, 1, 2, 5, 10)$ across columns, while we keep the kernel bandwidth in all models being 1.

## K  WHY WE CARE ABOUT THE ELBO

While we have shown that DIWP offers some benefits in predictive performance, it gives much more dramatic improvements in the ELBO. While we might think that predictive performance is the *only* goal, there are two reasons to believe that the ELBO itself is also an important metric. First, the ELBO is very closely related to PAC-Bayesian generalisation bounds (e.g. Germain et al., 2016). In particular, the bounds are generally written as the average training log-likelihood, plus the KL-divergence between the approximate posterior over parameters and the prior. This mirrors the standard form for the ELBO,

$$\mathcal{L} = \mathop{\mathbb{E}}_{Q(z)} \left[\log P\left(x|z\right)\right] - D_{\text{KL}}\left(Q\left(z\right) \| P\left(z\right)\right), \tag{102}$$

where $x$ is all the data (here, the inputs, $\mathbf{X}$ and outputs, $\mathbf{Y}$), and $z$ are all the latent variables. Remarkably, Germain et al. (e.g. 2016) present a bound on the test-log-likelihood that is exactly the ELBO per data point, up to additive constants. As such, in certain circumstances, optimizing the ELBO is equivalent to optimizing a PAC-Bayes bound on the test-log-likelihood. Similar results are available in Rivasplata et al. (2019). Second, we can write down an alternative form for the ELBO as the model evidence, minus the KL-divergence between the approximate and true posterior,

$$\mathcal{L} = \log P\left(x\right) - D_{\text{KL}}\left(Q\left(z\right) \| P\left(z|x\right)\right) \le \log P\left(x\right). \tag{103}$$

As such, for a fixed generative model, and hence a fixed value of the model evidence, $\log P\left(x\right)$, the ELBO measures the closeness of the variational approximate posterior, $Q\left(z\right)$ and the true posterior, $P\left(z|x\right)$. As we are trying to perform Bayesian inference, our goal should be to make the approximate posterior as close as possible to the true posterior. If, for instance, we can set $Q\left(z\right)$ to give better predictive performance, but be further from the true posterior, then that is fine in certain settings, but not when the goal is inference. Obviously, it is desirable for the true and approximate posterior to be as close as possible, which corresponds to larger values of $\mathcal{L}$ (indeed, when the approximate posterior equals the true posterior, the KL-divergence is zero, and $\mathcal{L} = \log P\left(x\right)$ ).

## L  DIFFERENCES WITH SHAH ET AL. (2014)

For a one-layer deep inverse Wishart process, using our definition in Eq. (16)

$$\mathbf{K}_0 = \tfrac{1}{N_0}\mathbf{X}\mathbf{X}^T, \tag{104a}$$

$$P\left(\mathbf{G}_1|\mathbf{K}_0\right) = \mathcal{W}^{-1}\left(\delta_1\mathbf{K}_0, \delta_1 + (P+1)\right), \tag{104b}$$

$$P\left(\mathbf{y}_\lambda|\mathbf{K}_1\right) = \mathcal{N}\left(y_\lambda; \mathbf{0}, \mathbf{K}\left(\mathbf{G}_1\right)\right). \tag{104c}$$

Importantly, we do the nonlinear kernel transformation *after* sampling the inverse Wishart, so the inverse-Wishart sample acts as a generalised lengthscale hyperparameter (App. B), and hence dramatically changes the function family.

In contrast, for Shah et al. (2014), the nonlinear kernel is computed *before*, the inverse Wishart is sampled, and the inverse Wishart sample is used directly as the covariance for the Gaussian,

$$\mathbf{K}_0 = \mathbf{K}\left(\tfrac{1}{N_0}\mathbf{X}\mathbf{X}^T\right), \tag{105a}$$

$$P\left(\mathbf{G}_1|\mathbf{K}_0\right) = \mathcal{W}^{-1}\left(\delta_1\mathbf{K}_0, \delta_1 + (P+1)\right), \tag{105b}$$

$$P\left(\mathbf{y}_\lambda|\mathbf{K}_1\right) = \mathcal{N}\left(y_\lambda; \mathbf{0}, \mathbf{G}_1\right). \tag{105c}$$

This difference in ordering, and in particular, the lack of a nonlinear kernel transformation between the inverse-Wishart and the output is why Shah et al. (2014) were able to find trivial results in their model (that it is equivalent to multiplying the covariance by a random scale).

