# OpenReview forum: "Deep Kernel Processes"
_ICLR.cc/2021/Conference — Reject_

### Official Review · AnonReviewer1 · 2020-10-19
**The paper proposed the deep kernel processes, which is a generalization of several previous deep kernel methods. Based on it, the authors develop a new variational inference method, and achieved superior performance for some popular datasets.**

**Rating:** 7
**Confidence:** 2

**Review:**

The paper proposes the deep kernel processes, which can generalize several existing deep kernel methods, including deep Gaussian processes and Bayesian neural networks. I'm not very familiar with the existing literature on the deep kernel models, so I could not properly assess the novelty in this work. However, the work seems significant to me. This new formulation allows the authors to avoid the feature space, and form the deep structure entirely on the kernel space. Combined with the new inference scheme, the authors showed the proposed method has performance gain over DGP and NNGP on many applications.

My main concern about this work is about its practicality. I am not entirely familiar with the current development in this area, but I believe DGP as formulated in this paper is not very popular and scalability might be one of the main reasons. Could the author address some of the limitation on this side? How does it compare to some of the more scalable methods (such as deep kernel learning where only the last layer is GP)? How is the tradeoff between accuracy and runtime for different number of inducing points?

---

> ### Author Response · Authors · 2020-11-13
> **Response**
>
> Surprisingly, the computational complexity of DKPs is better than that of the standard formulation of DGPs (as detailed in the new Sec. 5, DGPs are linear in the number of intermediate layer features, and this dependence does not arise in DKPs).  In fact, DKP computational complexity is equivalent to that of non-deep inducing GPs, which are much more widely used.  Indeed, non-deep inducing GPs have been used on some very large problems (e.g. [1,2]).  Further, recent work [3] has shown that by exploiting the equivalence between Bayesian neural networks and DGPs, we can scale even DGPs to e.g. ResNet like models for CIFAR-10, and we believe that similar approaches might be applicable to DKPs.
>
> The computational complexity of both deep kernel learning and DKPs is equivalent to shallow inducing GPs, so we might expect somewhat similar performance for these methods. However, it is important to note that deep kernel learning and DKPs aren't necessarily directly comparable: deep kernel learning maximizes over the NN parameters, whereas in the DKP we perform full Bayesian inference over all parameters at all layers. At the same time, deep kernel learning offers more flexibility in the neural network architecture (e.g. convolutions, batchnorm) that we are hoping to develop in future work for DKPs.
>
> We leave an in-depth investigation of the tradeoff between accuracy and number of inducing points to future work, in part because it will be highly dependent on the dataset and kernel.  Nonetheless:
> * In our experiments, we found good performance despite using a relatively small number of inducing points (100).
> * We expect the behaviour to be very similar to that of non-deep inducing GPs as in the limit of $\delta_\ell \rightarrow \infty$, our model is a shallow inducing GP, with a complex kernel, and the number of required inducing points for inducing GPs is well understood and behaved [4].
>
> [1] Hensman J, Fusi N, Lawrence ND. Gaussian processes for big data. UAI (2013).
>
> [2] Izmailov P, Novikov A, Kropotov D. Scalable gaussian processes with billions of inducing inputs via tensor train decomposition. AISTATS (2018).
>
> [3] Ober SW, Aitchison L. Global inducing point variational posteriors for Bayesian neural networks and deep Gaussian processes. arXiv:2005.08140.
>
> [4] Burt DR, Rasmussen CE, Van Der Wilk M. Rates of convergence for sparse variational Gaussian process regression. ICML (2019)

---

### Official Review · AnonReviewer2 · 2020-10-27
**Official Blind Review #2**

**Rating:** 6
**Confidence:** 3

**Review:**

---
##### Summary:

This paper proposes deep kernel processes (DKPs), which can be viewed as a specific kind of deep Gaussian processes where the kernel can be written as a function of the Gram matrix. The features in the intermediate layers are integrated out and the Gram matrix are Wishart distributed. A doubly stochastic variational inference method is proposed to learn DKPs. The idea looks novel to me. My major concern is about the writing.

---
##### Pros:

The paper is well-motivated and the idea is neat. For kernels that can be written as functions of the Gram matrix, DKP seems to be a better formulation than DGP. Due to difficulty in posterior inference, the authors propose to use inverse Wishart prior over Gram matrices. The use of inverse Wishart prior is explained and discussed in Appendix, which is satisfying. The authors also explain how the sparse variational inference method for DKP sidesteps the difficulties of capturing permutation and rotational symmetries in standard variational inference of BNNs or DGPs. The experimental results show the effectiveness of the proposed method.

---
##### Cons:

The paper is hard to read and understand. There are a lot of undefined or inconsistent notations, for example:
(1) In the beginning of the Background section, the data dimension is sometimes denoted as uppercase $P$, sometimes denoted as lowercase $p$.

(2) Following Eqn. (2), the inverse $\mathbf{\Sigma}^{-1}$ is not defined. What is $\mathbf{\Sigma}$? It seems that by $\mathbf{\Sigma}^{-1}$ the authors mean the inverse of $\mathbf{S}$, but that is a weird notation.

(3) In section 3, the notations $\mathbf{F}_\ell$ and $N_\ell$ are not explained. Although it can be inferred from the equations (4a-4c) or from the DGP literature, it is still obscure. $\mathbf{K}$ is sometimes used as a function that operates on the Gram matrix and sometimes used as the Kernel matrix, which need.

(4) In Eqn. (8), $\bm{G}_{\ell}-1$ should be $\bm{G}_{\ell-1}$.

Following that, the author mentioned “so $\mathbf{G}_{\ell+1}$ can be written as …” but used subscripts $\ell$ and $\ell-1$ in Eqn. (9).

(5) In Eqn. (11), it is not clear what $\mathbf{K}_{\mathrm{features}}$ means.

I just mentioned a few and there are many more. I suggest the authors refine the writing thoroughly.

---

> ### Author Response · Authors · 2020-11-13
> **Response**
>
> Thanks for your comments! We have addressed many clarity/notations issues during the review period (including a few you specified), and will continue doing until the end of the response period.  In response to your specific comments:
>  1. Fixed.
>  2. Fixed.
>  3. We have considerably clarified the definitions of terms in Eq. 4, including $\mathbf{F}_\ell$, $\mathbf{N}_\ell$ and noting that $\mathbf{K}(\cdot)$ is a function that takes a Gram matrix and returns a kernel matrix, whereas $\mathbf{K}_\ell$ is a (possibly random) variable representing a kernel matrix. (Any other notation we could think of would only have made things more confusing).
>  4. Fixed G_\ell-1 and changed G_{\ell+1} to G_\ell.  (The latex support appeared to break for some of these expressions).
>  5. We clarified that $\mathbf{K}_\text{features}(\mathbf{F}_\ell)$ is a function that takes a feature-matrix, $\mathbf{F}_\ell$ and returns the kernel matrix.  We included Eq. 11 to clarify how our $\mathbf{K}(\mathbf{G}_\ell)$, which is non-standard because it takes Gram-matrices as input relates to a more typical GP approach, which would involve computing the kernel from features.
>
> In addition, we have:
> * In Eq.4a, corrected K_\ell = K(G_{\ell-1})
> * Added dimension for $\mathbf{X}$ above Eq. 4a.
> * Conditioned on K_\ell rather than F_{\ell-1} in Eq. 4b.
> * Added dimension for $\mathbf{F}_\ell$ below Eq. 4c.
> * Added description of $\mathbf{y}_\lambda$ above Eq. 5.
> * Added layer index to $\Xi$ just above and below Eq. 9.
> * Defined $\mathbf{K}_\text{features}$ and $k$ below Eq. 11
> * Typo below Eq. 16 ("a full rank an").
> * Added dimensions for $\mathbf{X}$, $\mathbf{F_\ell}$, and $\mathbf{G}_\ell$ below Eq. 16c (but they can also be inferred from dimensions written around Eq. 4).
> * Added dimensions for $\mathbf{S}$ and $\mathbf{V}$ above Eq. 18a, with a mention of $\delta\mathbf{S}_0$.
> * Typo below Eq. 19 "are a $P \times P$ matrices"
> * Below Eq. 19 wrong transposes in $\mathbf{K}_0 = \mathbf{X}^T \mathbf{X}/N_0$.  Fixed to $\mathbf{K}_0 = \mathbf{X} \mathbf{X}^T / N_0$.
> * Below Eq. 19, specified that $\mathbf{V}_\ell$ are learned.
> * Added dimensions for $\mathbf{X}_i$, $\mathbf{X}_t$ above Eq. 20.
>
> Further changes in appendix:
> * Added dimensions for $\mathbf{K}$ above Eq.32.
> * Added descriptions for $N$ and $\delta$ below Eq.32.
> * Altered Appendix B so as to more closely match notation in main text.
> * Added dimensions for $\mathbf{F}_1$ and $\mathbf{X}$ above Eq.36.
> * Added dimensions and description for $\mathbf{\Omega}$ and $k_{param}$ below Eq.36.
> * Added dimensions for $\mathbf{W}$ in Eq.38.
> * Added a description for $\mathbf{w}_\lambda^\ell$ under Eq.41b.
> * Changed the notation for Dirac-delta function to $\delta_D$ to distinguish it from the $\delta$ parameter in Eq.45.
> * Added dimensions for $\mathbf{P}$ below Eq.54.
> * Added dimensions for $\mathbf{P}_\ell$ below Eq.58c.
> * Added dimensions for $\mathbf{U}_\ell$ above Eq.63.
> * Added dimensions for $\mathbf{G}_\text{ii}^\ell, \mathbf{\Psi}_\text{ii}, g_\text{it}^\ell$, and $\psi_\text{it}^\ell$ under Eq.98.

---

### Official Review · AnonReviewer3 · 2020-10-28
**see review**

**Rating:** 5
**Confidence:** 3

**Review:**

This paper proposes a prior distribution over covariance matrices of kernels which is defined as a sequential graphical model where each variable is Wishart distributed and its scale matrix is a non-linear transformation of its predecesor variable on the graph. The paper begins by considering a DGP with isotropic kernels across the layers and realizes that the Gram matrices are Wishart distributed. Based on this, the paper proposes to bypass the inference of the features and sample the Gram matrices directly from Wishart distributions. This insight, in addition to the layered structure of DGPs, gives rise to the proposed prior distribution. Furthermore, given the restrictions of the Wishart distribution for modelling covariance matrices of arbitrary size [1], as well as the conjugacy properties of the inverse Wishart distribution, the paper uses the inverse Wishart distribution instead. Doubly stochastic variational inference is proposed for approximating the posterior distribution which includes the use of inducing points thanks to the marginalization properties of the inverse Wishart distribution. The experimental contribution consists of a comparison against DGP and Neural Network GP on the UCI, MNIST and CIFAR-10 dataset.

In terms of clarity, the readability of the paper is negatively affected by being too broad with the results presented. It seems that the equivalence between GPs with Wishart processes as priors and DGPs, BNN could be a paper on its own. Thus, no topic is presented with the depth needed. In regards to significance, it is not clear from the paper, how extensive is the family of kernels that can be generated from the prior distribution, given that the proposed model is equivalent to DGPs only in the case of isotropic kernels. In that sense, it is not clear from the paper why this approach would be beneficial over DGPs.

[1] Amar Shah, Andrew Wilson, and Zoubin Ghahramani. Student-t processes as alternatives to Gaus-
sian processes. In Artificial intelligence and statistics, pp. 877–885, 2014.

---

> ### Author Response · Authors · 2020-11-10
> **Response**
>
> Thanks for your comments.  We agree that there is alot in the paper, and we had considered breaking it up.  The issue is that:
>  * If we just wrote a paper on the observation that deep Wishart process are equivalent to DGPs, we wouldn't be able to do any experiments, because the approximate posterior we'd like to use, the non-central Wishart, has an extremely  expensive probability density function (Appendix E).
>  * If we just wrote a paper on deep inverse Wishart processes observation, its importance wouldn't be clear, as it would be difficult to link to DGPs/BNNs.
>
> We are planning to write a fleshed-out follow up for JMLR, and we will extensively update the paper to improve clarity, including in response to the other reviewers.
>
> We have included extensive appendices, and hope that in combination with the main paper, they give sufficient detail  on all points.  If there are any areas you'd like us to elaborate a further on, we'd be happy to do that.
>
> The family of kernels is anything that can be written as a function of the Gram matrix.  This includes isotropic kernels (as you state), but also the dot-product kernel, the relu-kernel (Cho and Saul 2009), the cosine kernel and others.  This captures almost all kernels used in practice in the GP literature, but we agree that it isn't every kernel you could use in a DGP.
>
> There are two key advantages of DKPs over DGPs:
>   1. Primarily, DKPs allow for _much_ better posterior approximations than DGPs.  A DGP has rotational symmetries in  the true posterior (Appendix D2) which are not captured by standard approximate posteriors.  In contrast, Gram matrices don't have these symmetries, offering the possibility of much better posterior approximations.  In fact, we even believe that in certain cases, the DKP posterior might even be *unimodal*.  These improvements are reflected in the empirical results for the ELBO, which can be written as the model-evidence minus the KL-divergence   from the approximate to the true posterior.
>   2. DKPs allow us to optimize $\delta_\ell$ (Eq 16b).  This is analogous to a width parameter in a Bayesian neural network: as the width goes to infinity the Gram matrix at the next layer becomes deterministic and  we end up with an NNGP.  For smaller widths, the prior over representations is more flexible/variable.  Likewise, $\delta_\ell$ in a deep inverse Wishart process goes to infinity, the Gram matrix at the next layer becomes deterministic (and exactly the same as in a NNGP).  Critically, in a BNN, we can't optimize the width (because it is  fixed by the network architecture), and we definitely can't send the width to infinity (as GPU memory is fixed).  In  contrast, we are able to optimize the deep inverse Wishart process degrees of freedom, which allows us to control the flexibility of the prior over representations, and we are able to use very large degrees of freedom, if that is beneficial for modelling the data.

---

### Official Review · AnonReviewer4 · 2020-10-30
**good paper, but needs some corrections.**

**Rating:** 6
**Confidence:** 3

**Review:**

##########################################################################
Summary:

The manuscript proposes a deep kernel processes model, where gram matrices are transformed by non-linear kernel functions and assumed to follow Wishart distributions.

##########################################################################
Reasons for score:

I think that method provides a new approach to implement deep kernel processes (it is fully kernel-based compared to Deep GPs which is based on the feature based representation). The method has all required features to make kernel based methods practical: it is still an inductive learning method (enable to make predictions for unseen test points) and provides doubly stochastic inducing-point based variational inference (to scale the model to large data set). However, the manuscripts could be improved in some aspects (please read my comments below).

##########################################################################
Pros:

1.	I think that the method is carefully designed and equipped with all required features to make kernel based methods practical.
2.	The method includes important models (DGPs, BNNs, and infinite BNNs) as special cases.
3.	The method shows superior performance over DGPs and infinite BNNs on the benchmark data sets.

##########################################################################
Cons:

1.	Some parts are confusing or unclear. For example, in equation (19b) the dimensions of the matrices are not matched to each other: K(G_{l-1}) \in R^{N_0 X N_0} but V_l V_l^T in R^{PXP}. In addition, in equation (22) please provide the exact form of Q(\Omega, {G_{l}}^L_{l-2}, F_{L+1}|X_i). Equation 19 (a,b,c) do not really show how the method can reduce the computational complexity based on inducing points.

2.	The manuscript does not include computational (memory and time) complexities. I also think that the method should be compared to the previous approaches, including deep Gaussian processes with sparse approximation, in terms of these computational complexities.

#########################################################################

Minor comments:
I think that it would be better if the authors included more detailed descriptions what “task-dependent representation” in introduction.
In the sentence above equation (7), “the DKP as an autoregressive process (Fig. 2)”. I think that Fig. 2 is not about “autoregressive process”?

---

> ### Author Response · Authors · 2020-11-13
> **Response**
>
> We have clarified many details (see response to AnonReviewer2 below) and will continue to do so through the review period.  In response to your specific comments:
>
> $\mathbf{G}_\ell$ is $P \times P$, and $\mathbf{G}_1 = \mathbf{X} \mathbf{\Omega} \mathbf{X}^T$ is $P\times P$ because $\mathbf{X}$ is $P\times N_0$ ($\mathbf{\Omega}$ is $N_0 \times N_0$).  That said, we agree that the original manuscript was not clear as to the sizes of matrices.  We have considerably clarified matrix sizes at many points within the manuscript (see AnonReviewer2 below), by adding additional discussion of matrix sizes under Eq. 4 and Eq. 16 to address your specific point.
>
> We have considerably elaborated the exact form of Q(\Omega, {G_\ell}^L_{\ell=2}, F_{L+1}|X) (e.g. including the new Eq. 25 giving the precise expression for the inducing terms.  In general, we have considerably elaborated this section, and we hope it is now much clearer.
>
> We have added a substantial section on complexity (Sec. 5).  In the standard (non-DSVI) case, the complexity is $\mathcal{O}(P_t^3)$, whereas in the inducing case, it is $\mathcal{O}(P_i^3 + P_i^2 P_t)$.  Thus, by using a small number of inducing points, we are able to convert a cubic dependence on the number of input points into a linear dependence, which gives much improved scaling for DSVI. This is exactly the same scaling as a standard (non-deep) inducing GP.  But remarkably (detailed in that Section) it is better than standard DGPs, which additionally scale with the number of intermediate-layer features.
>
> Minor:
>  * We have replaced "task-dependent representation" with "they lack the capability to learn a top-layer representation, which is believed to be crucial for the effectiveness of deep methods"
>  * We have taken the reference to Fig. 2 out of the autoregressive section.

---

### Decision · Program_Chairs · 2021-01-07
**Final Decision**

**Decision:**

Reject

**Comment:**

The paper contributes to the literature of deep kernel learning by proposing a deep probabilistic model based on inverse Wishart distributions. This is an interesting addition to the literature, and the authors have provided some experimental evidence of the superiority of their method compared to state of the art on deep GPs and NNGPs. The major concerns with the paper, though, are related to the clarity of writing and its readability. Additionally, the experimental comparison is thin and lacks exploration for alternative configurations for DGPs and NNGPs.

---

> ### Author Response · Authors · 2021-01-17
> **Response**
>
> The paper received excellent scores of 7,6,6,5, with all but one reviewer recommending acceptance.
>
> Concerns about clarity were comprehensively addressed during the review period (see extensive reviewer responses below).
>
> Not one of the four reviewers raised concerns about the experiments.  Instead, the reviewers were of the view that "The experimental results show the effectiveness of the proposed method."  Further, as AnonReviewer3 pointed out, the paper already contains almost too many methodological contributions, so it was not possible to fit any more experiments into the limited space.